# Pliocene expansion of $C_4$ vegetation in the core monsoon zone on the Indian Peninsula

Ann G. Dunlea[1], Liviu Giosan[2], Yongsong Huang[3]

[1]Marine Chemistry & Geochemistry, Woods Hole Oceanographic Institution, Woods Hole, MA, 02543, USA
[2]Geology & Geophysics, Woods Hole Oceanographic Institution, Woods Hole, MA, 02543, USA
[3]Department of Earth, Environmental, and Planetary Sciences, Brown University, Providence, RI, 02912, USA

*Correspondence to*: Ann G. Dunlea (adunlea@whoi.edu)

**Abstract.** The expansion of $C_4$ vegetation during the Neogene was one of the largest reorganizations of Earth's terrestrial biome. Once thought to be globally synchronous in the late Miocene, site-specific studies have revealed differences in the timing of the expansion and suggest that local conditions play a substantial role. Here, we examine the expansion of $C_4$ vegetation on the Indian Peninsula since the late Miocene by constructing a ~6 million year paleorecord with marine sediment from the Bay of Bengal at Site U1445 drilled during International Ocean Discovery Program Expedition 353. Analyses of element concentrations indicate the marine sediment originates from the Mahanadi River in the Core Monsoon Zone (CMZ) of the Indian Peninsula. Hydrogen isotopes of the fatty acids of leaf waxes reveal an overall decrease in the CMZ precipitation since the late Miocene. Carbon isotopes of the leaf wax fatty acids suggest $C_4$ vegetation on the Indian Peninsula existed before the end of the Miocene, but expanded to even higher abundances during the mid-Pliocene to mid-Pleistocene (~3.5 to 1.5 million years ago). Similar to the CMZ on the Indian Peninsula, a Pliocene expansion or re-expansion has previously been observed in northwest Australia and in East Africa, suggesting that these tropical ecosystems surrounding the Indian Ocean remained highly sensitive to changes in hydroclimate after the initial spread of $C_4$ plants in late Miocene.

## 1. Introduction

The expansion of plants using the $C_4$ photosynthetic pathway is one of the most dramatic reorganizations of the global biome during the Neogene. A widespread late-Miocene expansion (8 to 6 Ma) is well documented and many studies have interpreted the broadly synchronous timing as ecosystems adapting to decreasing $pCO_2$ (e.g., Ehleringer et al., 1991; Ehleringer and Cerling, 1995; Cerling et al., 1993, 1997; Herbert et al., 2016). However, an increasing number of studies have shown that the timing, regional patterns, rate and drivers of $C_4$ grassland expansion were much more diverse and complex (An et al., 2005; Behrensmeyer et al., 2007; Huang et al., 2007; Edwards et al., 2010; Zhou et al., 2014). Along with low $pCO_2$, a $C_4$ photosynthetic pathway is better adapted to higher temperature, aridity, seasonality, and during disturbances such as flood, droughts, and fires (e.g., Edwards et al., 2010, and references therein). The interplay of these parameters varies amongst

regions. Resolving the precise timing and factors leading to major changes in vegetation demands site-specific studies (Strömberg et al., 2011; Zhou et al., 2014).

Our study provides a novel piece of the puzzle in unraveling the complexities of $C_4$ expansion by constructing a 6 million year (Myr) record of $C_4$ vegetation and aridity on the Indian Peninsula. The marine sediment record is from International Ocean Discovery Program (IODP) Site U1445 (17°44.72'N, 84°47.25'E; 2,503m water depth; Fig. 1A) drilled

in the Bay of Bengal (BoB) close to the mouths of the Mahanadi River. Lithologies at Site U1445 include calcareous fossils, biosilica, silt, and clays (including glauconite), and are overall described as hemipelagic sediment (Clemens et al., 2016). The Indian Monsoon dictates climate patterns in the Mahanadi River drainage basin: rainy summers, dry winters, and an annual reversal of wind direction (Gadgil, 2003; Sarkar et al., 2015). Highly sensitive to the seasonal changes, more than 80% of runoff from the Mahanadi River into the BoB occurs during the summer (Chakrapani and Subramanian, 1990).

Previous reconstructions of Neogene $C_4$ expansion in regions affected by the Indian Monsoon use deposits originating in the Himalayas and their piedmont regions (France-Lanord and Derry, 1994; Quade and Cerling, 1995; Quade et al., 1995; Cerling et al., 1997; Freeman and Colarusso, 2001; Sanyal et al., 2004; Behrensmeyer et al., 2007; Galy et al., 2010; Ghosh et al., 2017). The Mahanadi River drains a relatively low-elevation region of the Indian Peninsula distinct from the nearby mountain ranges (e.g., the Western Ghats, the Himalaya, Indo-Burman ranges Fig. 1, Xie et al., 2006). With minimal

orographic precipitation in the Mahanadi River basin, rainfall in this "Core Monsoon Zone" (CMZ) represents the mean behavior of the Indian Monsoon (Fig. 1; Ponton et al., 2012; Sarkar et al., 2015; Giosan et al., 2017,and references therein).

Although agriculture dominates present-day vegetation, models of seasonal climate predict the natural flora of the Mahanadi basin would be closed-canopy, moist deciduous forests and moist-to-dry woodlands with rare open spaces (Fig. 1C, Zorzi et al., 2015 and references therein). Today the region encompasses a range of $C_3$ and $C_4$ vegetation, but proxies and

models suggest that the plant communities are highly sensitive to glacial-interglacial changes with nearly all flora utilizing a $C_4$ pathway during the last glacial maximum (Galy et al., 2008; Phillips et al., 2014; Zorzi et al., 2015, and references therein). The behavior of vegetation in the CMZ over million-year timescales is unknown.

Here, we use inorganic bulk geochemical analyses to fingerprint the origin of sediment at Site U1445 to be from the Mahanadi River. Then we use bulk organic and compound-specific biomarkers at the same site, including carbon and hydrogen

isotope measurements of leaf wax fatty acids, to reconstruct the changes in $C_4$ vegetation and rainfall in the CMZ of the Indian Peninsula over the last ~6 Myr (Fig. 2).

## 2. Methods

Over million-year timescales, Site U1445 had a constant sedimentation rate (~115 ± 15 m/Myr; Clemens et al., 2016). We fit a locally weighted spline to biostratigraphic and magnetostratigraphic age constraints from Hole U1445A (Clemens et

al., 2016) using CLAM software in R (Blaauw, 2010) to estimate the age of our samples (Fig. A1). Our samples and the age constraints were from the same hole. Uncertainty on the ages is estimated to be less than ±0.2 Myr. Shipboard scientists

observed thin turbiditic sequences (~2-20cm thick) throughout Site U1445 and the expansion and dissociation of gas hydrates upon recovery that may muddle a higher-resolution record (Clemens et al., 2016). However, Site U1445 has fewer and smaller turbidite deposits relative to other sites drilled in this region and the records spanning million-year timescales are likely relatively undisturbed.

To determine sediment provenance, we measured major, trace, and rare earth element concentrations on 30 bulk sediment samples spanning 0 to 6 Myr including light and dark layers of sediment (Sec. 2.1.). To reconstruct hydrological and vegetation changes, we analyzed bulk organics (Sec. 2.2.) as well as compound-specific biomarkers (Sec. 2.3.) from 57 samples. We constructed the sampling plan to characterize the differences in organic content and isotope composition between the lighter and darker layers of sediment. As such, samples for organic and biomarker analysis were collected from Site U1445 in pairs, visually targeting relatively light and dark layers at similar depths to capture the variability range on shorter timescales while characterizing longer trends.

## 2.1. Inorganic analyses of bulk major, trace, rare earth element concentrations

The samples we analyzed for major, trace, and rare earth element concentrations were originally collected for moisture and density (MAD) measurements onboard the JOIDES Resolution during IODP Expedition 353. Each sample was collected with a 2 cm diameter plastic syringe that fits into the top of a 10 cm$^3$ glass vial, allowing for the vial to be completely filled with sediment (Clemens et al., 2016). The samples were dried in a convective oven at 105°C ± 5°C for 24 hours (Clemens et al., 2016). The remaining sample preparation, digestions, and analyses were conducted at Boston University and a detailed description of the analytical geochemical procedures are presented in Dunlea et al. (2015). In summary here, sediment samples were hand-powdered with an agate mortar and pestle. For major elements, sample powders were digested by flux fusion (Murray et al., 2000) and analyzed by inductively coupled plasma-emission spectrometry (ICP-ES). For analysis of trace and rare earth elements, sample powders were dissolved in a heated acid cocktail ($HNO_3$, HCl, and HF, with later additions of $HNO_3$ and $H_2O_2$ after samples were dried down) under clean-lab conditions and analyzed by inductively couple plasma-mass spectrometry (ICP-MS). Three separate digestions of a matrix-matched in-house sediment standard were analyzed with each batch and determined precision [(standard deviation)/(average) x 100] was ~2% of the measured value for each element. The international Standard Reference Material BHVO-2 was analyzed as an unknown with each batch and results were consistently found to be accurate within precision for each element.

## 2.2. Analyses of carbon and nitrogen content and isotopes

Analyses of the abundance of total carbon (TC), total inorganic carbon (TIC), total organic carbon (TOC), nitrogen (N), and the δ$^{13}$C of the TOC component were performed at Woods Hole Oceanographic Institution and methods are described in Whiteside et al. (2011). In brief here, samples for TOC were weighed into tared silver boats and then acidified to remove carbonates in a closed desiccator for 3 days at 60-65°C over concentrated hydrochloric acid. All samples were flash combusted in a Costech 4010 Elemental Analyzer coupled via a Finnigan-MAT Conflo-II interface to a Thermo DeltaVPlus isotope ratio

mass spectrometer. Data were recorded and integrated using the Isodat software package. Post-run calculations were performed for blank corrections, quantifications, and final calibrations.

## 2.3. Analyses of compound specific biomarkers abundances and isotopes

The analyses of compound-specific biomarkers were performed at Brown University (e.g., Daniels et al., 2017). Samples were freeze-dried and lipids were extracted from 3.5 to 4.5 g of sediment using a Dionex 350 Accelerated Solvent Extractor (ASE) with dichloromethane:methanol (9:1 v/v). The fatty acids in the total lipid extract were separated from the neutral lipids using aminopropyl silica gel chromatography, eluting with a dichloromethane:isopropanol solution followed by ether with 5% acetic acid.

The fatty acids were methalyated to form fatty acid methyl ester (FAME) by dissolving dried down acid fraction in in ~0.3 mL of toluene and ~1mL of 5:95 acetyl chloride:methanol with known isotopic composition. Nitrogen replaced the headspace in the vial before they were capped tightly and heated at 60°C for 12 hours. Once the reaction was complete, the FAMEs were separated from the water by-products formed during the methylation reaction. Sample received ~1mL of synthetic saline solution (50g NaCl/L of double-distilled water) and ~1mL of hexane, were vigorously shaken, and then allowed to rest until the hexane separated from the water. The hexane fraction was pipetted into a new vial, avoiding the water. Another ~1mL of hexane was added to the sample, shaken, and pipetted into the new vial. To clean the solution and isolate the fatty acids, samples were run through a second silica gel column, eluting with hexane to remove unwanted acids and then DCM to acquire the clean FAME fraction.

The FAME fraction was analyzed on an Agilent 6890 gas chromatograph with a flame ionization detector (GC-FID). Sample blanks were analyzed with every batch. The isotope ratios of the FAME fraction ($\delta D_{n\text{-acid}}$ and $\delta^{13}C_{n\text{-acid}}$) were measured on a Thermo Finnigan Delta + XL isotope ratio mass spectrometer with a HP 6890 gas chromatograph and a high-temperature pyrolysis reactor for sample introduction. For δD, three injections of each sample were analyzed and two injections of each sample were analyzed for $\delta^{13}C$. Between every six injections, an in-house lab standard mixture containing known amounts of various n-acids was analyzed to monitor instrument accuracy and precision. The δD and $\delta^{13}C$ were corrected for the methyl groups added during derivatization using the following equations $\delta D_{corrected}=[(2n+2)*\delta D_{measured}+123.7*3]/(2n-1)$ and $\delta^{13}C_{corrected} =[(n+1)*\delta^{13}C_{measured}+ 36.52]/n$, where n is the carbon chain length of the compound and the methanol added has δD=-123.6‰ and $\delta^{13}C$=-36.52‰. Analytical uncertainty was calculated by [standard deviation/average] of the injections and is typically less than 3% for δD and less than 1% for $\delta^{13}C$. The standard deviations are reported in Table S3. For every instrument run, samples were analyzed in random order.

## 2.4. Correction of plant physiological effects on δD

We corrected the δD for differences in $C_3$ versus $C_4$ plant physiology (Fig. E1; Smith and Freeman, 2006; Chikaraishi and Naraoka, 2007). First, we calculated the fraction of $C_3$ versus $C_4$ vegetation with $\delta^{13}C_{FA}$ for each sample, estimating that $C_3$ and $C_4$ vegetation have a $\delta^{13}C_{FA}$ of -37.7 ± 1.8 ‰ and -21.1 ± 1.4 ‰, respectively (Chikaraishi et al., 2004; Ponton et al.,

2012). Then we approximate that $C_3$ plants have a δD that is 30 ‰ lighter than $C_4$ plants (Smith and Freeman, 2006; Chikaraishi and Naraoka, 2007; Ghosh et al., 2017) and use the fraction of $C_3$ versus $C_4$ to correct the δD of each sample for the differences in plant physiology (Fig. E1). The corrected data shows the same overall trend as the uncorrected data, except the values are shifted to be more negative and the overall change in δD is steeper (Fig. E1).

## 3. Results

To determine the provenance of the aluminosilicate fraction, we examined the proportions of Al, Ti, Sc, Nb, La, and Th concentrations, because other elements (e.g., Fe, K, Mg, Si, Zr, Hf) may be affected by continental weathering, sorting during transport, and post-depositional authigenic processes. The results from 30 samples have almost constant element proportions of the selected elements, indicating that the aluminosilicate fraction of sediment did not significantly vary over

135 the past 6 Myr. The composition of the 30 samples, even amongst the light and dark layers, matches the composition of lithologies that comprise the Mahanadi basin such as Precambrian granite and gneisses of the Indian craton and associated sedimentary deposits (Sharma, 2009; Fig. B1; Table S1). Marine sediment deposits in other parts of the Bay of Bengal closer to the Krishna and Godavari Rivers or Ganges-Brahmaputra Rivers have a more mafic or highly variable composition that is not observed at Site U1445 (e.g., Tripathy et al., 2014; Fig. B1). As such, we interpret our results as recording terrestrial

changes in the CMZ, specifically the Mahanadi drainage basin.

The pairs of samples used for organic analyses are spaced ~28 m apart (~260 kyr intervals) and the adjacent light and dark layers within each pair were 0.2 m to 4.3 m apart in the sediment core (2 kyr to 46 kyr; Fig. A1). The color difference can be related to the total organic carbon content (wt%; TOC) and total nitrogen content with darker layers having 1.0 to 2.8 times more than adjacent lighter layers (Fig. 2A; Fig. C1; Table S2).

Long-chain normal fatty acids of leaf waxes are derived from land plants and are well preserved during transport and burial in marine sediment (Eglinton and Eglinton, 2008). We focus on the $C_{30}$ chain length to avoid possible contaminations from non-terrestrial sources that contribute shorter chain length fatty acids (Fig. D1; Table S3). The results of the $δ^{13}C$ of $C_{30}$ fatty acid of leaf waxes ($δ^{13}C_{FA}$) show a 5‰ increase from mid-Pliocene to mid-Pleistocene (~3.5 to ~1.5 Ma), after which $δ^{13}C_{FA}$ decreases and becomes more variable from ~1.5 Ma to the present (Fig. 2B). The hydrogen isotope compositions of the

leaf wax fatty acids ($δD_{FA}$) increase gradually over the past 6 Myr, but have a wide range amongst light and dark layers and shorter time intervals (Fig. 2C; Table S3). Comparing the earlier and later time intervals when $δ^{13}C_{FA}$ changes, before the mid-Pliocene (3.5 Ma) $δD_{FA}$ ranges from -179‰ to -147‰ and after the mid-Pleistocene (1.5 Ma) the $δD_{FA}$ increases to between -166‰ to -126‰ (Fig. 2C).

## 4. Discussion

### 4.1 C$_4$ Expansion on the Indian Peninsula

The $\delta^{13}C_{FA}$ of terrestrial plants is primarily a function of the photosynthetic pathway and isotopic composition of atmospheric $CO_2$ (e.g., Farquhar et al., 1989). In this study, the 5‰ increase in $\delta^{13}C_{FA}$ is greater than the reconstructed $\delta^{13}C$ of atmospheric $CO_2$ ($\leq$1‰; Tipple et al., 2010), suggesting that a correction for $\delta^{13}C_{CO2}$ would only slightly adjust our results. Thus we interpret $\delta^{13}C_{FA}$ as reflecting the amount of C$_4$ relative to C$_3$ vegetation produced in the CMZ.

From ~6 Ma until ~3.5 Ma, approximately 51% to 81% (avg. 69% ± 9% s.d.) of the vegetation in the CMZ utilized a C$_4$ photosynthetic pathway (Fig. E1). Thus, the environmental threshold for C$_4$ photosynthetic pathway had already been crossed before the end of the late Miocene. Later in the mid-Pliocene, the reconstruction shows another distinct expansion reaching 64% to 92% (avg. 81% ± 7% s.d.) of C$_4$ vegetation in the early Pleistocene (Fig. 2B). The change in vegetation from ~3.5 to ~1.5 Ma suggests multiple steps of C$_4$ expansion in the CMZ, rather than a singular late-Miocene expansion. From ~1.5 Ma to the present, the average proportion of C$_4$ vegetation decreased and became more variable (58% to 92%, avg. 76% ± 10% s.d.; Fig. 2B), which may reflect the sensitivity of the region to glacial-interglacial variations observed in shorter records from this region (e.g., Zorzi et al., 2015, and references therein).

### 4.2 Aridification of the Indian Peninsula

After correcting for the effects of plant physiology (Sec. 2.4.), the amount of precipitation and mixing of different air masses can each vary the hydrogen isotopic composition of leaf wax fatty acids ($\delta D_{FA}$; e.g., Eglinton and Eglinton, 2008). The mixing of two air masses with unique $\delta D$ values was recently observed to drive $\delta D$ of rainfall in New Delhi, India, but, similar to the amount of precipitation, the relatively depleted $\delta D$ corresponded with wetter conditions (Hein et al., 2017). Thus, we interpreted the $\delta D_{FA}$ as a qualitative proxy for aridity or the relative amount of precipitation. The $\delta D_{FA}$ results suggest an overall drying of the CMZ on the Indian Peninsula over the past 6 Myr. The shorter-term scatter in the $\delta D_{FA}$ record may reflect higher frequency variations in aridity or rainfall.

### 4.3 Patterns of C$_4$ Expansion surrounding the Indian Ocean

In this section, we compare our record of C$_4$ expansion with other compound-specific biomarker records of C$_4$ expansion at sites in the Indian Ocean or adjacent land and seas. Multiple proxy records document a late-Miocene C$_4$ expansion in the Ganges or Brahmaputra River basins such as the Siwalik Group in northern Pakistan or BoB sites receiving outflow sediment (Fig. 3A France-Lanord and Derry, 1994; Quade and Cerling, 1995; Cerling et al., 1997; Freeman and Colarusso, 2001; Sanyal et al., 2004; Behrensmeyer et al., 2007; Ghosh et al., 2017). Collectively, the reported timing of C$_4$ expansions in the Himalaya region ranges from 9 to 5 Myr, most commonly 8 to 6 Myr (Behrensmeyer et al., 2007). Rather than a uniform timing, detailed sampling of various deposits around the Siwalik regions shows that C$_4$ vegetation expansion was staggered amongst nearby sub-environments with different local conditions (Ghosh et al., 2017, and references therein). Another

biomarker record documents a late-Miocene expansion in a wide continental region north and west of the Arabian Sea (Site 722; Fig. 1A, Fig. 3B; Huang et al., 2007). Once $C_4$ vegetation expanded at each of these sites, the records suggest there is overall little or no systematic change in the amount of $C_4$ vegetation after the late Miocene.

In contrast, the CMZ of the Indian Peninsula and a few other records around the Indian Ocean document an expansion of $C_4$ vegetation during the Pliocene (Fig. 3). Marine deposits in the Gulf of Aden originate from northeast Africa and record

a late-Miocene $C_4$ expansion, followed by a relapse to predominantly $C_3$ vegetation ~4.3 Ma and a re-expansion of $C_4$ plants in the Pliocene (Site 231; Fig. 1A, Fig. 3C; Feakins et al., 2005, 2013; Liddy et al., 2016). The Pliocene re-expansion is consistent with other records from tropical East Africa (e.g., Levin et al., 2004; Cerling et al., 2011). A $C_4$ expansion in the Pliocene is also observed in northwest Australia (Site 763A; Fig. 3E; Andrae et al., 2018). There is little evidence of significant $C_4$ vegetation prior to the Pliocene, suggesting a relatively late onset of $C_4$ vegetation expansion in northwest Australia (Fig.

3). Additionally, there is evidence that East, South, and Central Asia experienced multiple steps of $C_4$ expansion through the Pliocene (e.g., An et al., 2005; Passey et al., 2009; Zhou et al., 2014; Miao et al., 2017; Koutsodendris et al., 2019). Collectively, a significant regional expansion in the Pliocene, distinctly after the first late-Miocene expansion, is common at least amongst tropical East Africa, Northwest Australia, Asia, and the Indian Peninsula.

### 4.4 Triggers of $C_4$ Expansion in the Pliocene

The adaptations of the $C_4$ photosynthetic pathway provides a competitive advantage over $C_3$ vegetation in environmental conditions with low $pCO_2$, high temperature, high aridity or extreme seasonality that can lead to frequent floods, droughts, and fires (e.g., Edwards et al., 2010, and references therein). Many studies hypothesize that the global expansion of $C_4$ vegetation in the late Miocene was triggered by $pCO_2$ decreasing below a temperature-dependent threshold (Herbert et al., 2016 and references therein). During the Pliocene, decreasing $pCO_2$ may have also contributed to the expansion of $C_4$

vegetation (Fig. 3a; Pagani et al., 2009; Tripati et al., 2009; Seki et al., 2010; Bartoli et al., 2011). However, the heterogenous regional response of the Pliocene $C_4$ expansion suggests $pCO_2$ cannot be the only driver. Given that the Pliocene $C_4$ expansion is observed in multiple regions surrounding the Indian Ocean that have highly seasonal rainfall, regional hydrodynamics likely played a role in the Pliocene $C_4$ expansion.

In the modern era, a complex interplay of multiple modes of variability dictate the unique seasonal and multi-decadal

precipitation patterns in the regions surrounding the Indian Ocean. For example, rainfall on the Indian Peninsula follows quintessential monsoon behavior, but is also tied to the Inter-Tropical Convergence Zone (ITCZ), Walker Circulation, and the Indian Ocean Dipole (IOD; Gadgil, 2003; Wang et al., 2017). The biannual rains of the (semi)arid tropical East Africa are related to the ITCZ, monsoon winds, sea surface temperature (SST), and Walker Circulation (e.g., Williams and Funk, 2011; Tierney et al., 2015; Yang et al., 2015). Monsoon rains annually quench northern Australia, but the El Niño Southern

Oscillation (ENSO), Walker Circulation, and the amount of Indonesian Throughflow (ITF) better explain the precipitation in other parts of Australia (Ummenhofer et al., 2009, 2011a, 2011b). Previous studies reason that changes in monsoon precipitation or frequency of fires likely triggered local expansion of $C_4$ vegetation (An et al., 2005; Passey et al., 2009; Zhou

et al., 2014; Miao et al., 2017; Andrae et al., 2018). However, the sampling resolution of our million-year study cannot resolve these seasonal, decadal, or centennial rhythms in precipitation, so we examine the underlying processes driving variations in each of these modes of variability.

The physical mechanism of each of these modes of variability can be related back to atmospheric pressure gradients. Monsoons are classically defined as a seasonal wind reversal induced by the pressure gradient caused by differential heating of land and sea (Wang et al., 2017). Broadly, the ITCZ is formed by the convergence and uplift of air masses near the equator due to differential heating between low and high latitudes and the formation of Hadley Cells. The IOD and ENSO are cyclic changes in the zonal gradient across the Indian and Pacific Oceans, respectively, and affect Walker circulation. Variable Walker circulation affects ocean upwelling and thermocline depth, which in turn will influence sea surface temperature. Sea surface temperatures can interact with the atmosphere and reinforce or dampen the atmospheric pressure gradients affiliated with the monsoon, ITCZ, IOD, or ENSO dynamics. Changes in any combination of these atmospheric gradients may have altered the seasonal precipitation in the regions surrounding the Indian Ocean during the Pliocene (Wang et al., 2017).

What could have changed atmospheric pressure gradients in the Pliocene? Indonesian Throughflow was being restricted from 5 to 3 Myr and there is evidence for strengthening SST gradients during the Pliocene (Cane and Molnar, 2001; Wara et al., 2005; Karas et al., 2009; Ford et al., 2012; Zhang et al., 2014; Burls and Fedorov, 2017; Christensen et al., 2017; White and Ravelo, 2020). Changes in the zonal and meridional SST gradients would have affected atmospheric pressure gradients in and surrounding the Indian Ocean and modified seasonal precipitation patterns. At the same time, the onset of northern hemisphere glaciation likely also helped reorganize the atmospheric pressure gradients surrounding the Indian Ocean (e.g., Koutsodendris et al., 2019). Before the intensification of glaciation at 2.7 Ma, there is evidence of four early local glaciation events at 4.9–4.8 Myr, 4.0 Myr, 3.6 Myr and 3.3 Myr (De Schepper et al., 2014). The latter two events are within age model uncertainty of 3.5 Myr when $C_4$ vegetation expands in various regions around the Indian Ocean. Thus, it is possible that the early local growth of ice, not the intensification of glaciation, altered the Siberian high or other atmospheric pressure gradients near the Indian Ocean changing cyclic precipitation patterns that lead to a regional expansion in $C_4$ vegetation.

On the Indian Peninsula, $C_4$ vegetation expands gradually from 3.5 Ma until the Mid-Pleistocene when the range of variability in $C_4$ vegetation increases. The timing of the change coincides within uncertainty with the Mid-Pleistocene transition (~1.3 Ma) where the glacial periods transitioned from 41 kyr cycles to 100 kyr cycles. Although the trigger for the switch in cyclicity is unknown, the processes responsible for the change or the ice sheets themselves may have affected atmospheric gradients and the modes of climate variability that altered the dynamics of $C_4$ expansion on the Indian Peninsula. The sampling resolution from 1.5-0 Ma in our study is too low to resolve the change in cyclicity. However, previous studies documenting variations in $C_4$ vegetation relative to $C_3$ since the last glacial maximum demonstrate the sensitivity of plant communities to recent glacial-interglacial cycles (Galy et al., 2008; Phillips et al., 2014; Zorzi et al., 2015, and references therein).

## 5. Conclusion

Our study provides a piece of the puzzle in unraveling the complexities of $C_4$ vegetation expansion. Although $C_4$ vegetation was established in the CMZ on the Indian Peninsula before the end of the Miocene, the results of this study show another significant expansion in the Pliocene (~3.5 to 1.5 Myr). The latter expansion is not observed in many records from the orographically-wet Himalaya emphasizing the spatial heterogeneities in $C_4$ vegetation response – even within the same monsoon system. However, other regions adjacent to the Indian Ocean, such as tropical East Africa, Asia, and Northwest Australia, corroborate the observed expansion in the CMZ of the Indian Peninsula and show $C_4$ vegetation patterns sensitive to the changes in hydroclimate during the Pliocene. The heterogeneous response suggests that $pCO_2$ cannot be the exclusive driver of the expansion of the $C_4$ vegetation in the Pliocene and changes in regional hydrodynamics likely contributed. Restriction of Indonesian Throughflow and the onset on Northern Hemisphere glaciation may have altered the atmospheric pressure gradients and the modes of variability that determine seasonal precipitation patterns in the continents surrounding the Indian Ocean, which may have caused the regional expansion of $C_4$ vegetation in the Pliocene.

**Appendices**

**Appendix A. Figure A1. Age-depth model for Site U1445. To determine the ages of our samples, we fit the biostratigraphic and magentostratigraphic age constraints (Clemens et al., 2016) with an age-depth model using CLAM software in R (Blaauw, 2010). We ran iterations of the model with different types of fit and levels of smoothing, and identified a locally weighted spline with 0.4 smoothing to best represent the trends observed in the age constraints. The differences between the age models iterations are not significant and would not change the interpretations of this study.**

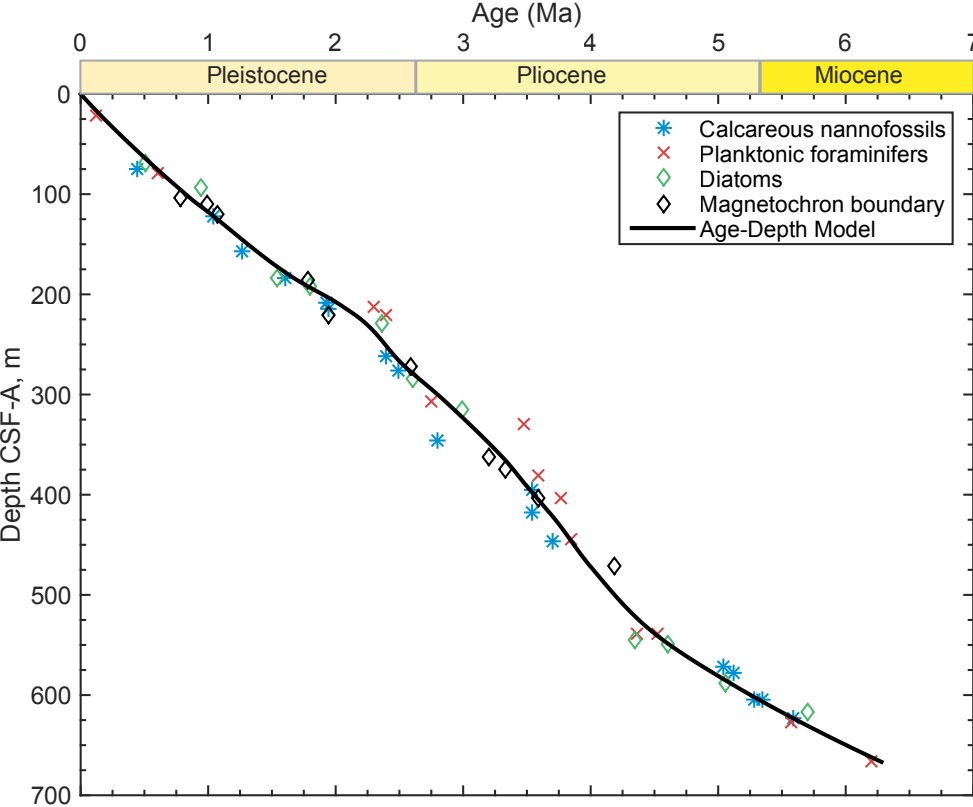

**Appendix B. Figure B1. La-Th-Sc diagram of 30 sediment samples from the Bay of Bengal. Samples from IODP Site U1445 (blue squares) are plotted as well IODP Site U1444 (green diamonds), NGHP Site 19 (purple triangles), and NGHP Site 16 (brown circles) in the Bay of Bengal. Average upper continental crust (black square, Rudnick and Gao, 2014), post-Archean average Australian Shale (black dot, Taylor and McLennan, 1985), and average mid-ocean ridge basalt (Gale et al., 2013) compositions are plotted for reference.**

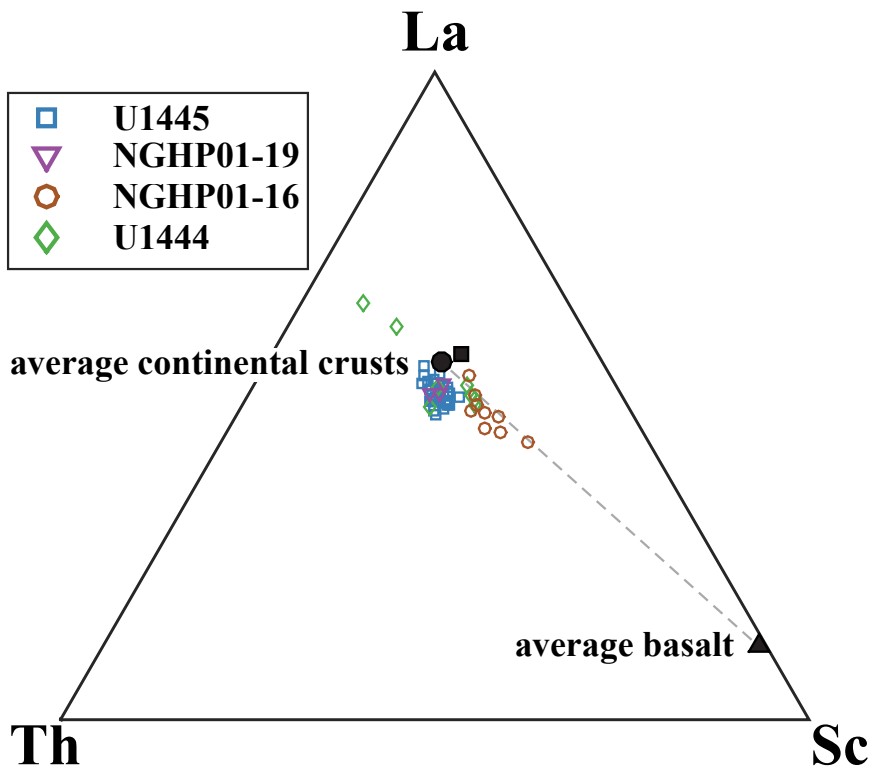

**Appendix C. Figure C1. Carbonate and bulk organic analyses at Site U1445. Analysis of 57 samples at IODP Site U1445 for (a) bulk calcium carbonate content (weight %) calculated as (total inorganic carbon x (8.33313 CaCO₃ wt/C wt)), (b) total organic carbon concentration (weight %), (c) total acidified nitrogen content (weight %), (d) Ratio of total organic carbon to total nitrogen (TOC/TN, wt.%/wt.%) and (e) carbon isotopes of the total organic carbon (per mil). Black dots represent visually darker layers relative to a lighter layer (white dot) at a similar depth. TOC/TN shows a distinct increase in the mid-Pliocene, but remains within the range of TOC/TN expected for marine organic material.**

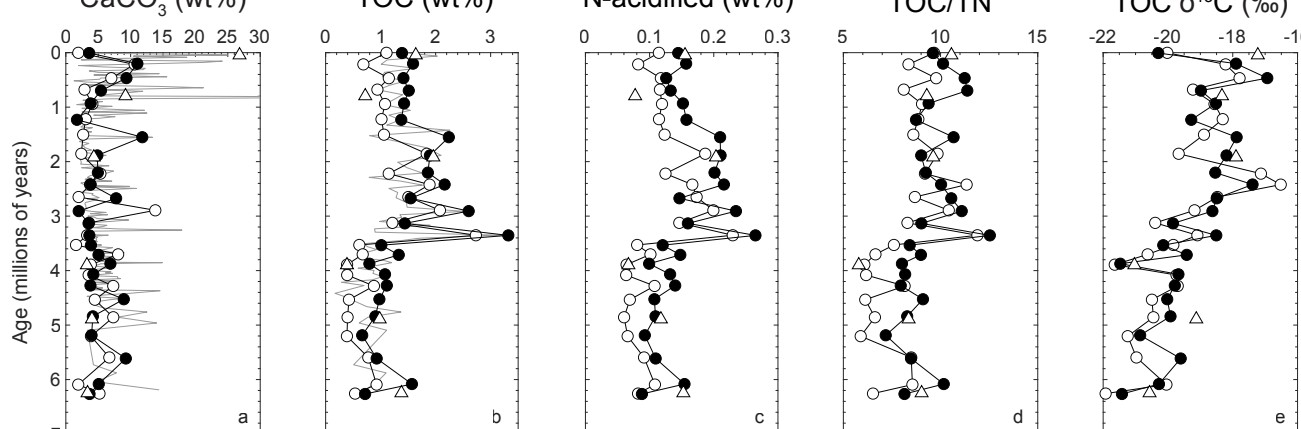

**Appendix D. Figure D1. Long-chain fatty acids from leaf waxes extracted from Site U1445. Plotted from left to right are hydrogen isotopes and then carbon isotopes of leaf wax fatty acids from chainlengths C₂₆, C₂₈, and C₃₀.**

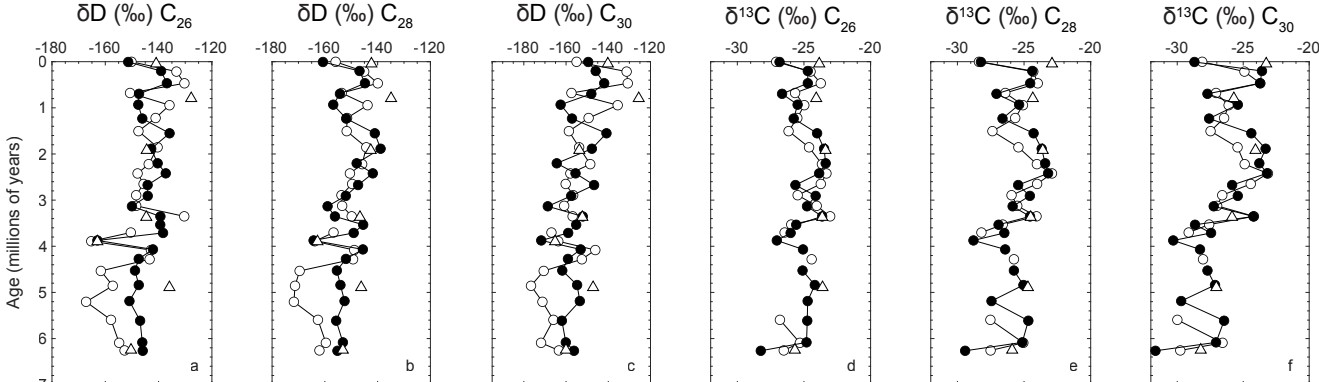

**Appendix E. Figure E1. Correcting δD for differences in fractionation due to plant physiology. (a) Raw, uncorrected δ¹³C_FA plotted for comparison. (b) Calculated fraction of C₄ vegetation with grey lines indicating the maximum and minimum boundaries. (c) Uncorrected δD data, plotted here for comparison. (d) The δD data corrected for differences between C₃ and C₄ plant physiologies.**

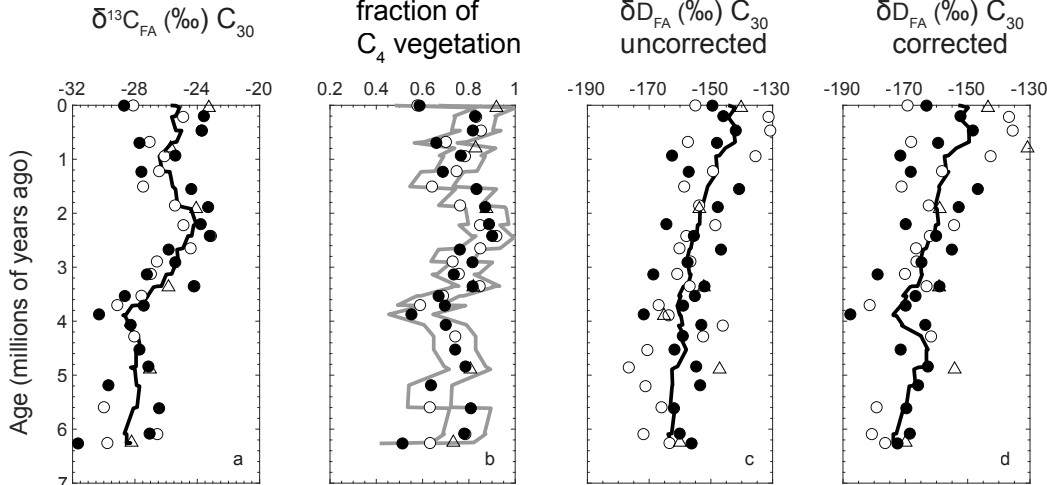

## Data Availability

Data is included in the supplementary tables and is publicly available in the PANGAEA Database (https://www.pangaea.de/).

## Author Contribution

A.G.D. participated in the IODP Expedition 353 sampling party, performed the geochemical analyses, and lead writing and revisions of the manuscript. L.G. participated on IODP Expedition 353, was involved in the project's conceptualization, sample acquisition, and provided supervision. Y.H. also participated on IODP Expedition 353, was involved in the project's conceptualization, provided resources and funding acquisition, advised on the methodology, supervision, and aided in interpretation of data.

## Competing Interests

The authors declare that they have no conflict of interest.

## Acknowledgements

We thank Raj Kumar Singh (IIT Bhubaneswar, India) for providing Mahanadi sediment samples, X. Wang, R. Tarozo, and M. Da Rosa Alexandre at Brown Univ. and T. Ireland at Boston Univ. for their analytical assistance and as well as S. Clemens, K. Thirumalai, V. Galy, and C. Ummenhofer for discussions and advice. This research used samples and data provided by the International Ocean Discovery Program. Funding for this research was provided by the Ocean and Climate Change Institute Postdoctoral Scholarship at Woods Hole Oceanographic Institution to AGD, and the U.S. National Science Foundation to LG (NSF OCE-0652315). USSSP post-cruise support was provided to Exp. 353 shipboard participants LG and YH.

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

**Figures**

Figure 1. (a) Location of IODP Site U1445 in the Bay of Bengal (red star). Site 231 in the Gulf of Aden and Site 722 in the Arabian Sea are plotted for reference (red dots). Topography and bathymetry are represented in the background map. The Mahanadi River and main tributaries are traced in dark blue and the region outlined by the box is zoomed-in for Figures 1a and 1c, which are modified from Ponton et al. (2012). (b) average annual rainfall (m/year) and (c) natural ecosystems in the region including the Mahanadi River drainage basin.

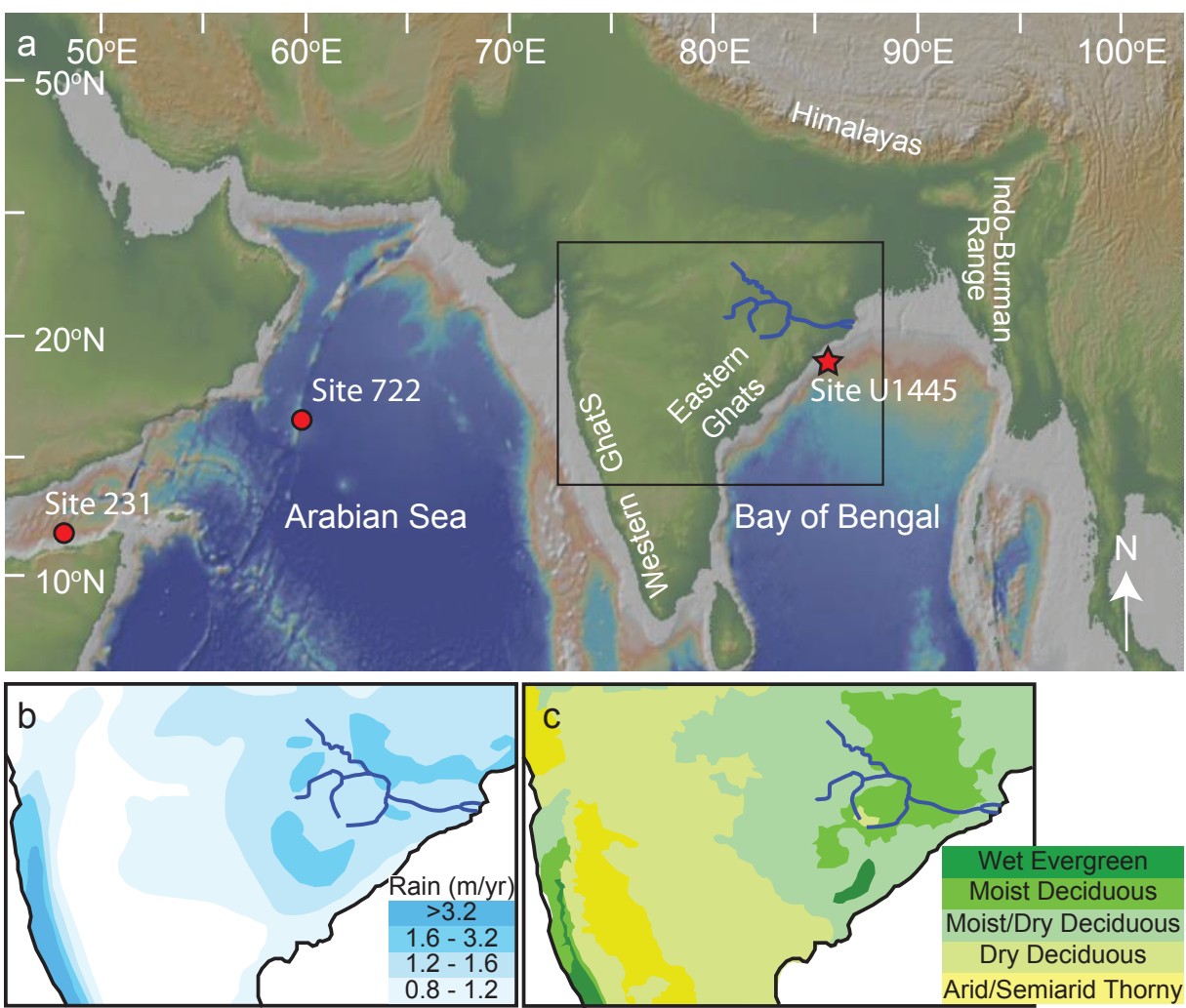

Figure 2. Analyses of 57 samples from Site U1445 in the Bay of Bengal plotted with age (Ma) and (a) total organic carbon (wt. %).), (b) carbon isotope values of $C_{30}$ fatty acids from leaf waxes ($\delta^{13}C_{FA}$, per mil), and (c) hydrogen isotope values of $C_{30}$ fatty acids from leaf waxes ($\delta D_{FA}$, per mil). Black and white dots are pairs of samples from relatively dark and light layers, respectively, at a similar depth. Triangles are samples not in pairs. Black curves are a 9-point moving average of all samples.

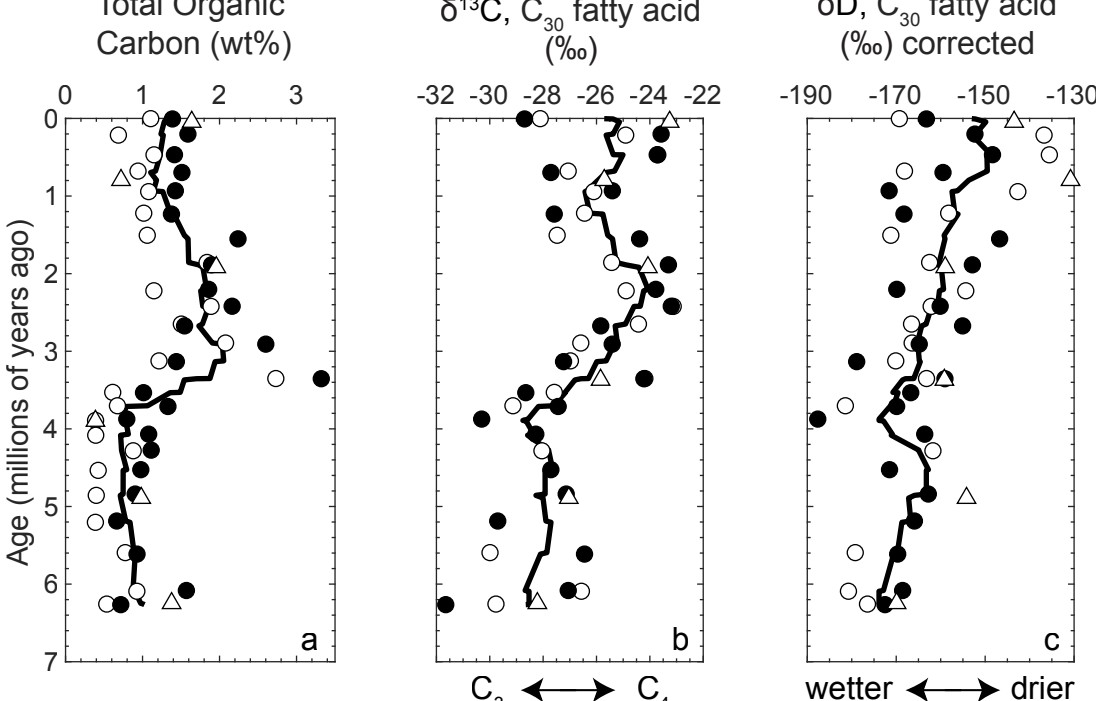

**Figure 3. (a)** Proxy estimates of $pCO_2$ over the past 6 Myr (green, $\delta^{11}B$, Bartoli et al., 2011; red, alkenone, Pagani et al., 2009; blue, B/Ca Tripati et al. (2009); yellow, alkenone, Seki et al., 2010; purple, $\delta^{11}B$, Seki et al., 2010). **(b)** LR04 global stack of benthic $\delta^{18}O$ records as a proxy for global ice volume (Lisiecki and Raymo, 2005) with arrows marking early glaciation events (De Schepper et al., 2014). Blue and red highlight values above and below modern $\delta^{18}O$, respectively. **(c)** $\delta^{13}C$ of $C_{33}$-alkanes from Siwalik paleosols in Northern Pakistan (white dots) and from sediment at Site 717 in the Bengal Fan (black dots; Freeman and Colarusso, 2001). **(d)** $\delta^{13}C$ of $C_{31}$-alkanes at Site 722 in the Arabian Sea (Huang et al., 2007), which integrates vegetation variability from north and east of the Arabian Sea. **(e)** $\delta^{13}C$ of $C_{28}$-fatty acids at Site 231 in the Gulf of Aden, which records vegetation in East Africa (Feakins et al., 2013; Liddy et al., 2016). **(f)** $\delta^{13}C$ of $C_{30}$-fatty acid at Site U1445 in the Bay of Bengal, which records vegetation from the Mahanadi basin on the Indian Peninsula (this study). **(g)** $\delta^{13}C$ of $C_{33}$-alkanes from northwest Australia (Andrae et al., 2018).

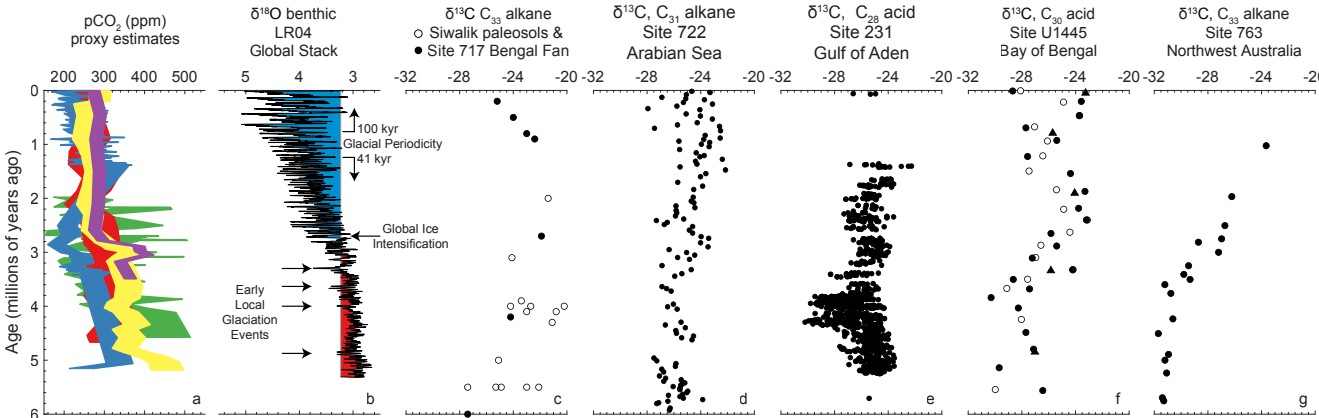

**Supplemental Material**

**Table S1. Inorganic analyses of major, trace, and rare earth element concentrations for 30 bulk sediment samples at Site U1445 and additional samples from other sites in the Bay of Bengal for reference. For methods see Appendix A or details in Dunlea et al. (2015).**

 **Table S2. Analyses of 57 bulk sediment samples from Site U1445 for bulk calcium carbonate, total organic carbon, total carbon, total acidified nitrogen, carbon isotopes of the total organic carbon, and the designation of visually lighter versus darker samples at similar depths.**

**Table S3. Hydrogen isotopes and carbon isotope analyses of leaf wax fatty acids extracted from 57 samples at Site U1445. Measurements from fatty acid chainlengths $C_{26}$, $C_{28}$, and $C_{30}$ are reported with their standard deviation. The correction for $C_3$-$C_4$ physiological differences in the hydrogen isotopes of $C_{30}$ fatty acids is reported, estimating $C_3$ vegetation as having a $\delta^{13}C$ of -35.4 ‰**
 **and $C_4$ vegetation as -21.4‰.**

**Supplemental Table S1**

| Exp | Site | Hole | Core | Type | Sect | W | Top | Bot | Depth mbsf, CSF-A | Age Ma | Si wt. % | Al wt. % | Ti wt. % | Fe wt. % | Mn wt. % | Ca wt. % | Mg wt. % | Na wt. % | K wt. % | P wt. % | Li ppm | Be ppm | Sc ppm | V ppm | Cr ppm | Co ppm | Ni ppm |
|---|---|---|---|---|---|---|---|---|---|---|---|---|---|---|---|---|---|---|---|---|---|---|---|---|---|---|---|
| Mahanadi River | S3 | Bulk | | | | | | | | | 29.0 | 7.0 | 0.6 | 3.6 | 0.0 | 0.7 | 0.5 | 0.5 | 2.2 | 0.0 | 29.5 | 2.8 | 15.6 | 101.8 | 100.3 | 13.3 | 43.5 |
| Mahanadi River | S3 | Coarse Grain Size | | | | | | | | | 31.5 | 5.7 | 0.4 | 2.7 | 0.1 | 0.7 | 0.3 | 0.9 | 2.8 | 0.0 | 17.0 | 2.1 | 8.6 | 58.6 | 46.7 | 17.8 | 25.6 |
| NGHP Exp. 01 | 19 | A | 33 | X | CC | | 16 | 24 | 253.56 | 6-10Ma | 25.3 | 9.3 | 0.5 | 4.9 | 0.0 | 0.9 | 1.5 | 1.4 | 2.6 | 0.1 | 80.8 | 3.7 | 17.5 | 137.9 | 136.9 | 17.8 | 67.6 |
| NGHP Exp. 01 | 19 | A | 36 | X | CC | | 34 | 51 | 282.34 | 6-10Ma | 24.7 | 8.3 | 0.4 | 4.0 | 0.0 | 2.7 | 1.3 | 1.4 | 2.3 | 0.1 | 72.9 | 3.1 | 16.9 | 129.8 | 140.3 | 16.3 | 65.7 |
| NGHP Exp. 01 | 19 | A | 38 | X | CC | | 22 | 33 | 301.82 | 6-10Ma | 25.1 | 9.3 | 0.5 | 4.8 | 0.0 | 1.3 | 1.5 | 1.4 | 2.5 | 0.0 | 84.4 | 3.5 | 17.9 | 137.0 | 138.7 | 16.4 | 64.6 |
| NGHP Exp. 01 | 16 | A | 1 | H | 1 | | 38 | 40 | 0.38 | | 21.4 | 8.1 | 0.7 | 7.1 | 0.1 | 1.9 | 1.9 | 2.4 | 1.8 | 0.1 | 49.5 | 2.1 | 24.4 | 189.3 | 154.4 | 28.8 | 77.1 |
| NGHP Exp. 01 | 16 | A | 1 | H | 4 | | 29 | 31 | 4.79 | | 22.2 | 8.6 | 0.6 | 6.2 | 0.0 | 1.4 | 1.8 | 2.1 | 2.1 | 0.1 | 62.3 | 2.5 | 20.9 | 138.0 | 151.8 | 22.0 | 73.8 |
| NGHP Exp. 01 | 16 | A | 2 | H | 2 | | 4 | 6 | 8.54 | | 16.7 | 6.0 | 0.4 | 4.3 | 0.0 | 11.2 | 1.6 | 1.7 | 1.6 | 0.1 | 44.6 | 1.8 | 15.0 | 104.9 | 112.3 | 17.2 | 62.4 |
| NGHP Exp. 01 | 16 | A | 2 | H | 3 | | 6 | 8 | 10.06 | | 18.5 | 6.7 | 0.5 | 5.2 | 0.0 | 6.5 | 1.6 | 1.9 | 1.8 | 0.1 | 53.9 | 2.0 | 18.9 | 140.2 | 138.6 | 20.2 | 73.0 |
| NGHP Exp. 01 | 16 | A | 2 | H | 4 | | 98 | 100 | 12.48 | | 18.4 | 7.0 | 0.5 | 5.7 | 0.0 | 6.3 | 1.6 | 1.8 | 1.8 | 0.1 | 59.8 | 2.1 | 17.1 | 124.9 | 136.4 | 18.7 | 75.9 |
| NGHP Exp. 01 | 16 | A | 2 | H | 6 | | 48 | 50 | 14.98 | | 17.7 | 6.6 | 0.5 | 4.8 | 0.0 | 9.1 | 1.5 | 1.6 | 1.8 | 0.1 | 49.5 | 2.0 | 15.3 | 122.7 | 120.2 | 19.0 | 64.1 |
| NGHP Exp. 01 | 16 | A | 3 | H | 2 | | 4 | 6 | 18.04 | | 21.4 | 8.3 | 0.6 | 6.0 | 0.0 | 2.8 | 1.7 | 2.0 | 2.0 | 0.1 | 74.2 | 2.3 | 21.7 | 158.1 | 159.2 | 21.4 | 74.8 |
| NGHP Exp. 01 | 16 | A | 3 | H | 4 | | 6 | 8 | 21.06 | | 21.1 | 8.0 | 0.5 | 6.5 | 0.0 | 3.7 | 1.6 | 1.8 | 2.1 | 0.1 | 67.3 | 2.4 | 19.1 | 141.9 | 127.4 | 21.3 | 72.8 |
| NGHP Exp. 01 | 16 | A | 3 | H | 7 | | 49 | 51 | 25.49 | | 22.2 | 8.4 | 0.7 | 6.3 | 0.0 | 2.3 | 1.8 | 1.7 | 2.2 | 0.1 | 62.7 | 2.5 | 21.4 | 179.0 | 152.9 | 26.4 | 73.3 |
| 353 | U1444 | A | 6 | H | 4 | W | 108 | 110 | 51 | 0.2 | 33.0 | 6.6 | 0.4 | 3.1 | 0.1 | 2.0 | 1.2 | 1.7 | 2.3 | 0.1 | 31.6 | 3.0 | 10.7 | 68.5 | 67.0 | 10.9 | 30.4 |
| 353 | U1444 | A | 12 | H | 3 | W | 60 | 62 | 98 | 0.4 | 19.3 | 7.4 | 0.4 | 5.1 | 0.4 | 7.5 | 1.6 | 1.4 | 2.0 | 0.1 | 63.2 | 2.6 | 18.6 | 122.6 | 131.5 | 43.1 | 146.1 |
| 353 | U1444 | A | 15 | X | 4 | W | 81 | 83 | 123 | 1.2 | 22.1 | 8.3 | 0.5 | 5.0 | 0.2 | 5.5 | 1.7 | 1.3 | 2.3 | 0.1 | 80.1 | 2.7 | 17.9 | 153.9 | 141.0 | 31.1 | 123.2 |
| 353 | U1444 | A | 17 | X | 4 | W | 14 | 16 | 142 | 2.6 | 23.8 | 7.5 | 0.4 | 4.4 | 0.1 | 4.7 | 1.5 | 1.2 | 2.3 | 0.0 | 72.8 | 3.4 | 14.5 | 159.0 | 133.6 | 31.3 | 156.6 |
| 353 | U1444 | A | 19 | X | 4 | W | 20 | 22 | 161 | 3.5 | 22.1 | 8.8 | 0.5 | 5.2 | 0.1 | 4.1 | 1.7 | 1.5 | 2.1 | 0.1 | 85.7 | 2.9 | 18.3 | 166.3 | 151.2 | 29.7 | 113.2 |
| 353 | U1444 | A | 24 | F | 3 | W | 51 | 53 | 208 | 3.65 | 34.7 | 5.0 | 0.4 | 2.8 | 0.1 | 2.1 | 0.9 | 1.6 | 1.8 | 0.1 | 24.8 | 2.6 | 9.3 | 52.7 | 46.9 | 7.6 | 19.6 |
| 353 | U1444 | A | 31 | X | 3 | W | 74 | 76 | 267 | 3.75 | 26.2 | 10.0 | 0.5 | 4.9 | 0.1 | 0.9 | 1.9 | 1.3 | 3.4 | 0.1 | 82.3 | 4.3 | 17.2 | 148.4 | 131.4 | 18.7 | 52.9 |
| 353 | U1444 | A | 33 | X | 4 | W | 90 | 92 | 288 | 5.0 | 18.9 | 7.4 | 0.4 | 4.3 | 0.1 | 10.4 | 1.2 | 1.2 | 1.7 | 0.0 | 70.8 | 2.5 | 16.5 | 121.9 | 116.7 | 28.0 | 89.5 |
| 353 | U1445 | A | 1 | H | 5 | W | 23 | 25 | 6.23 | 0.04 | 19.7 | 6.9 | 0.4 | 4.1 | 0.1 | 8.0 | 1.9 | 1.9 | 2.1 | 0.1 | 56.4 | 2.4 | 13.0 | 112.0 | 100.4 | 24.0 | 69.0 |
| 353 | U1445 | A | 2 | H | 6 | W | 40 | 42 | 14.8 | 0.11 | 21.5 | 8.4 | 0.4 | 5.2 | 0.1 | 2.9 | 1.8 | 3.0 | 2.3 | 0.1 | 76.8 | 3.0 | 15.7 | 109.9 | 108.2 | 22.0 | 70.7 |
| 353 | U1445 | A | 3 | H | 4 | W | 30 | 32 | 21.2 | 0.16 | 17.1 | 6.4 | 0.4 | 3.9 | 0.1 | 11.1 | 1.6 | 1.4 | 2.1 | 0.1 | 46.0 | 2.1 | 12.3 | 98.4 | 97.1 | 25.7 | 66.3 |
| 353 | U1445 | A | 4 | H | 4 | W | 115 | 117 | 31.57 | 0.24 | 24.4 | 9.3 | 0.5 | 5.3 | 0.1 | 1.2 | 1.7 | 1.8 | 2.7 | 0.0 | 89.9 | 3.3 | 19.5 | 145.2 | 123.6 | 23.4 | 77.3 |
| 353 | U1445 | A | 5 | H | 4 | W | 105 | 107 | 40.86 | 0.31 | 21.3 | 7.7 | 0.4 | 4.6 | 0.2 | 4.9 | 1.8 | 1.7 | 2.3 | 0.1 | 66.7 | 2.5 | 16.3 | 121.1 | 116.6 | 27.2 | 82.0 |
| 353 | U1445 | A | 6 | H | 4 | W | 77 | 79 | 50.02 | 0.39 | 21.4 | 7.4 | 0.5 | 4.3 | 0.2 | 6.7 | 1.7 | 1.5 | 2.4 | 0.1 | 58.7 | 2.4 | 14.1 | 128.3 | 101.5 | 38.3 | 82.0 |
| 353 | U1445 | A | 7 | H | 4 | W | 98 | 100 | 59.49 | 0.47 | 22.5 | 8.8 | 0.5 | 5.0 | 0.1 | 2.6 | 1.8 | 1.5 | 2.4 | 0.1 | 78.9 | 2.9 | 18.8 | 148.5 | 125.2 | 26.8 | 89.0 |
| 353 | U1445 | A | 9 | H | 4 | W | 110 | 112 | 77.54 | 0.62 | 21.9 | 8.2 | 0.4 | 5.0 | 0.1 | 4.8 | 1.8 | 1.4 | 2.4 | 0.1 | 67.8 | 2.6 | 17.3 | 137.4 | 118.8 | 27.4 | 97.3 |
| 353 | U1445 | A | 11 | H | 4 | W | 61 | 63 | 96.82 | 0.79 | 23.5 | 9.0 | 0.5 | 5.2 | 0.1 | 3.0 | 1.7 | 1.5 | 2.7 | 0.1 | 84.5 | 3.3 | 18.2 | 144.3 | 114.9 | 22.2 | 74.5 |
| 353 | U1445 | A | 13 | H | 4 | W | 69 | 71 | 116.31 | 0.98 | 23.7 | 9.0 | 0.5 | 5.0 | 0.1 | 2.4 | 1.7 | 1.6 | 2.5 | 0.0 | 81.0 | 3.0 | 18.1 | 132.0 | 116.9 | 21.4 | 72.6 |
| 353 | U1445 | A | 15 | H | 4 | W | 76.5 | 78.5 | 135.755 | 1.17 | 23.3 | 8.5 | 0.5 | 5.3 | 0.1 | 2.7 | 1.8 | 1.8 | 2.5 | 0.1 | 79.4 | 2.9 | 17.3 | 143.7 | 112.1 | 32.7 | 102.9 |
| 353 | U1445 | A | 17 | H | 4 | W | 133 | 135 | 155.24 | 1.36 | 24.3 | 9.0 | 0.5 | 4.9 | 0.1 | 1.0 | 1.9 | 2.1 | 2.6 | 0.1 | 89.7 | 3.2 | 18.7 | 153.4 | 121.5 | 24.6 | 90.0 |
| 353 | U1445 | A | 19 | H | 4 | W | 78 | 80 | 173.35 | 1.55 | 21.6 | 7.4 | 0.4 | 4.5 | 0.1 | 4.8 | 1.7 | 1.6 | 2.0 | 0.1 | 75.6 | 2.5 | 15.7 | 110.9 | 118.8 | 20.4 | 75.4 |
| 353 | U1445 | A | 20 | H | 4 | W | 90 | 92 | 183.13 | 1.66 | 24.1 | 8.7 | 0.5 | 5.3 | 0.1 | 1.5 | 1.7 | 1.6 | 2.4 | 0.1 | 87.7 | 2.9 | 19.4 | 143.9 | 128.0 | 24.0 | 88.5 |
| 353 | U1445 | A | 21 | H | 4 | W | 43 | 45 | 192.36 | 1.79 | 24.0 | 8.8 | 0.5 | 5.0 | 0.1 | 1.6 | 1.7 | 1.7 | 2.5 | 0.1 | 86.4 | 2.9 | 17.5 | 129.9 | 116.7 | 24.0 | 74.8 |
| 353 | U1445 | A | 22 | H | 4 | W | 88 | 90 | 201.6 | 1.92 | 24.4 | 8.1 | 0.4 | 4.8 | 0.1 | 1.4 | 1.6 | 1.7 | 2.3 | 0.1 | 80.8 | 2.7 | 17.0 | 116.2 | 110.9 | 20.2 | 79.8 |
| 353 | U1445 | A | 24 | H | 4 | W | 35 | 37 | 219.72 | 2.14 | 22.7 | 9.1 | 0.5 | 5.4 | 0.1 | 1.5 | 1.6 | 2.0 | 2.4 | 0.1 | 92.1 | 3.1 | 18.4 | 141.5 | 125.4 | 32.4 | 96.0 |
| 353 | U1445 | A | 25 | X | 4 | W | 38 | 40 | 229.21 | 2.24 | 23.5 | 9.6 | 0.5 | 5.3 | 0.1 | 1.2 | 1.7 | 1.2 | 2.4 | 0.1 | 95.9 | 3.1 | 19.4 | 146.4 | 124.2 | 19.4 | 79.3 |
| 353 | U1445 | A | 28 | X | 5 | W | 94 | 96 | 260.1 | 2.45 | 23.9 | 9.6 | 0.5 | 5.0 | 0.1 | 0.6 | 1.6 | 1.6 | 2.4 | 0.0 | 105.8 | 3.4 | 19.3 | 147.7 | 130.1 | 24.3 | 100.6 |
| 353 | U1445 | A | 31 | X | 4 | W | 43 | 45 | 287.37 | 2.68 | 25.7 | 7.9 | 0.4 | 4.6 | 0.1 | 1.1 | 1.4 | 1.9 | 2.2 | 0.0 | 81.4 | 3.0 | 16.3 | 121.2 | 118.4 | 25.0 | 79.3 |
| 353 | U1445 | A | 34 | X | 4 | W | 66 | 68 | 316 | 2.93 | 23.2 | 8.8 | 0.5 | 4.6 | 0.1 | 2.6 | 1.7 | 1.8 | 2.3 | 0.1 | 88.6 | 3.1 | 18.6 | 151.6 | 120.0 | 27.2 | 81.7 |
| 353 | U1445 | A | 37 | X | 4 | W | 35 | 37 | 338 | 3.12 | 23.4 | 8.5 | 0.5 | 5.4 | 0.1 | 1.8 | 1.5 | 1.7 | 2.1 | 0.1 | 89.5 | 2.9 | 17.4 | 147.2 | 131.9 | 24.0 | 97.4 |
| 353 | U1445 | A | 41 | X | 2 | W | 7 | 9 | 369 | 3.35 | 25.2 | 7.5 | 0.4 | 4.6 | 0.1 | 2.1 | 1.4 | 1.6 | 2.2 | 0.1 | 80.2 | 3.1 | 16.1 | 121.4 | 112.0 | 19.1 | 93.1 |
| 353 | U1445 | A | 46 | X | 4 | W | 11 | 13 | 412 | 3.64 | 25.3 | 9.2 | 0.5 | 4.8 | 0.1 | 1.1 | 1.6 | 1.7 | 2.7 | 0.0 | 93.1 | 3.7 | 18.3 | 137.6 | 114.8 | 18.6 | 62.5 |
| 353 | U1445 | A | 52 | X | 4 | W | 53 | 55 | 460 | 3.93 | 25.2 | 9.9 | 0.5 | 4.8 | 0.2 | 0.9 | 1.7 | 1.3 | 3.2 | 0.0 | 84.6 | 4.0 | 18.8 | 144.1 | 117.4 | 22.5 | 57.8 |
| 353 | U1445 | A | 56 | X | 4 | W | 94 | 96 | 493 | 4.14 | 24.9 | 9.1 | 0.5 | 4.7 | 0.1 | 1.8 | 1.5 | 2.0 | 2.7 | 0.0 | 91.3 | 3.2 | 17.1 | 132.9 | 120.5 | 19.6 | 71.4 |
| 353 | U1445 | A | 61 | X | 4 | W | 78 | 80 | 533 | 4.45 | 25.0 | 9.2 | 0.4 | 4.6 | 0.1 | 2.0 | 1.7 | 1.3 | 3.1 | 0.0 | 82.3 | 3.9 | 17.2 | 132.9 | 104.4 | 21.5 | 66.6 |
| 353 | U1445 | A | 66 | X | 4 | W | 57 | 59 | 573 | 4.89 | 24.2 | 8.4 | 0.4 | 5.0 | 0.1 | 2.6 | 1.4 | 1.3 | 2.0 | 0.1 | 83.0 | 3.0 | 18.4 | 151.7 | 124.0 | 22.7 | 88.4 |
| 353 | U1445 | A | 71 | X | 4 | W | 132 | 134 | 620 | 5.54 | 25.5 | 8.4 | 0.4 | 5.1 | 0.1 | 1.6 | 1.2 | 1.3 | 2.0 | 0.0 | 84.1 | 2.9 | 17.2 | 133.8 | 112.2 | 23.6 | 98.5 |
| 353 | U1445 | A | 75 | X | 4 | W | 83 | 85 | 655 | 6.09 | 24.8 | 8.5 | 0.4 | 4.9 | 0.1 | 1.7 | 1.5 | 1.4 | 2.2 | 0.0 | 76.7 | 3.2 | 18.5 | 154.6 | 128.9 | 19.8 | 78.4 |

| Cu | Zn | Rb | Sr | Y | Zr | Nb | Mo | Sn | Sb | Cs | Ba | La | Ce | Pr | Nd | Sm | Eu | Gd | Tb | Dy | Ho | Er | Yb | Lu | Hf | Ta | Pb | Th | U |
|---|---|---|---|---|---|---|---|---|---|---|---|---|---|---|---|---|---|---|---|---|---|---|---|---|---|---|---|---|---|
| ppm | ppm | ppm | ppm | ppm | ppm | ppm | ppm | ppm | ppm | ppm | ppm | ppm | ppm | ppm | ppm | ppm | ppm | ppm | ppm | ppm | ppm | ppm | ppm | ppm | ppm | ppm | ppm | ppm | ppm |
| 34.0 | 64.8 | 142.2 | 121.5 | 52.3 | 164.5 | 22.9 | 0.5 | 3.3 | 0.5 | 5.8 | 729.6 | 67.6 | 127.1 | 13.9 | 50.0 | 9.8 | 1.9 | 8.4 | 1.2 | 7.1 | 1.3 | 3.6 | 3.5 | 0.5 | 4.3 | 2.5 | 29.9 | 35.9 | 3.7 |
| 20.5 | 42.0 | 149.9 | 152.5 | 19.7 | 131.0 | 14.9 | 0.5 | 2.0 | 0.3 | 3.8 | 946.9 | 37.5 | 73.1 | 7.1 | 25.8 | 5.0 | 1.3 | 4.2 | 0.6 | 3.6 | 0.7 | 1.9 | 1.9 | 0.3 | 3.2 | 1.0 | 29.7 | 16.1 | 2.0 |
| 56.4 | 119.2 | 169.2 | 124.5 | 24.9 | 78.4 | 16.4 | 1.4 | 4.4 | 1.2 | 12.7 | 532.3 | 36.7 | 77.5 | 8.1 | 29.5 | 5.9 | 1.2 | 5.2 | 0.8 | 4.4 | 0.9 | 2.4 | 2.3 | 0.4 | 2.4 | 1.1 | 30.9 | 18.6 | 3.6 |
| 58.5 | 117.6 | 147.5 | 195.0 | 25.8 | 71.5 | 14.9 | 1.0 | 3.6 | 1.0 | 10.5 | 439.8 | 34.7 | 69.7 | 7.9 | 28.7 | 5.8 | 1.2 | 5.2 | 0.8 | 4.5 | 0.9 | 2.5 | 2.4 | 0.4 | 2.3 | 1.0 | 25.7 | 15.3 | 4.5 |
| 50.5 | 111.0 | 145.1 | 130.5 | 23.1 | 70.4 | 15.5 | 1.6 | 4.2 | 1.0 | 12.1 | 515.1 | 35.3 | 72.7 | 8.0 | 29.2 | 5.8 | 1.2 | 4.8 | 0.7 | 4.1 | 0.8 | 2.2 | 2.2 | 0.3 | 2.4 | 1.1 | 27.3 | 16.9 | 4.0 |
| 97.7 | 95.3 | 91.2 | 116.6 | 23.7 | 93.2 | 13.8 | 1.0 | 2.7 | 0.8 | 5.4 | 199.5 | 25.5 | 55.8 | 6.2 | 23.6 | 5.2 | 1.2 | 4.9 | 0.8 | 4.4 | 0.9 | 2.3 | 2.2 | 0.3 | 3.0 | 0.9 | 15.2 | 9.7 | 2.6 |
| 65.9 | 99.4 | 112.0 | 123.1 | 20.2 | 85.6 | 14.7 | 1.0 | 3.0 | 0.9 | 6.8 | 270.1 | 27.3 | 58.6 | 6.4 | 23.7 | 5.0 | 1.1 | 4.4 | 0.7 | 3.7 | 0.8 | 2.0 | 1.9 | 0.3 | 2.5 | 0.9 | 18.2 | 12.7 | 3.9 |
| 51.5 | 76.6 | 86.6 | 74.5 | 20.5 | 90.0 | 13.0 | 1.0 | 2.0 | 0.9 | 5.0 | 280.9 | 24.7 | 48.9 | 5.7 | 21.1 | 4.3 | 0.9 | 3.9 | 0.6 | 3.3 | 0.7 | 1.8 | 1.7 | 0.3 | 2.0 | 0.7 | 13.5 | 9.6 | 5.5 |
| 68.9 | 94.5 | 98.3 | 390.1 | 20.4 | 94.1 | 12.4 | 2.0 | 2.3 | 1.3 | 5.7 | 307.2 | 25.1 | 48.9 | 5.8 | 21.5 | 4.5 | 1.0 | 4.2 | 0.6 | 3.6 | 0.7 | 2.0 | 1.9 | 0.3 | 2.6 | 0.8 | 15.0 | 9.7 | 5.7 |
| 59.9 | 89.0 | 104.0 | 351.0 | 18.1 | 78.0 | 12.3 | 4.2 | 2.3 | 1.3 | 6.0 | 370.0 | 24.6 | 50.2 | 5.6 | 20.4 | 4.2 | 0.9 | 3.8 | 0.6 | 3.3 | 0.7 | 1.8 | 1.7 | 0.3 | 2.1 | 0.8 | 15.8 | 10.1 | 5.8 |
| 54.7 | 78.1 | 94.3 | 1028.5 | 18.9 | 80.6 | 12.7 | 0.9 | 2.5 | 0.9 | 5.7 | 286.9 | 29.2 | 54.8 | 6.4 | 23.2 | 4.6 | 1.0 | 4.1 | 0.6 | 3.4 | 0.7 | 1.8 | 1.7 | 0.3 | 2.2 | 0.8 | 15.9 | 10.3 | 4.8 |
| 74.4 | 102.5 | 110.7 | 189.1 | 19.6 | 90.2 | 13.3 | 0.9 | 2.7 | 0.9 | 6.5 | 343.5 | 26.4 | 55.4 | 6.2 | 22.7 | 4.7 | 1.1 | 4.3 | 0.7 | 3.7 | 0.8 | 2.0 | 1.9 | 0.3 | 2.5 | 0.8 | 17.7 | 11.2 | 5.1 |
| 63.2 | 96.2 | 121.6 | 228.4 | 20.2 | 84.4 | 14.0 | 1.5 | 2.9 | 1.1 | 7.0 | 405.5 | 29.2 | 59.9 | 6.7 | 24.7 | 5.0 | 1.1 | 4.4 | 0.7 | 3.8 | 0.8 | 2.0 | 1.9 | 0.3 | 2.4 | 0.9 | 18.9 | 13.1 | 5.0 |
| 81.6 | 107.4 | 119.4 | 162.7 | 22.2 | 87.6 | 15.8 | 1.0 | 3.2 | 0.9 | 6.7 | 312.8 | 33.2 | 68.6 | 7.5 | 27.5 | 5.6 | 1.2 | 5.0 | 0.8 | 4.2 | 0.8 | 2.2 | 2.0 | 0.3 | 2.7 | 1.0 | 19.5 | 13.7 | 3.7 |
| 17.3 | 55.6 | 138.5 | 171.6 | 25.6 | 22.2 | 14.0 | 0.4 | 4.1 | 0.4 | 6.8 | 406.7 | 44.3 | 86.4 | 9.8 | 35.3 | 7.0 | 1.2 | 5.8 | 0.9 | 4.7 | 0.9 | 2.4 | 2.2 | 0.3 | 0.7 | 1.1 | 20.8 | 18.0 | 2.3 |
| 130.5 | 140.2 | 120.4 | 403.4 | 25.8 | 74.0 | 11.9 | 0.9 | 2.8 | 1.9 | 7.8 | 857.1 | 30.0 | 65.9 | 7.0 | 25.8 | 5.5 | 1.2 | 5.0 | 0.8 | 4.3 | 0.9 | 2.4 | 2.3 | 0.4 | 2.0 | 0.8 | 30.4 | 12.0 | 2.3 |
| 108.4 | 146.1 | 134.6 | 307.9 | 21.7 | 69.6 | 12.6 | 0.6 | 3.3 | 1.5 | 8.8 | 846.5 | 29.9 | 61.5 | 6.9 | 25.1 | 5.1 | 1.1 | 4.6 | 0.7 | 3.9 | 0.8 | 2.1 | 2.1 | 0.3 | 1.9 | 0.9 | 23.8 | 12.0 | 2.6 |
| 87.3 | 138.7 | 140.4 | 283.5 | 24.9 | 82.4 | 14.2 | 1.2 | 3.5 | 2.1 | 9.2 | 735.4 | 30.0 | 75.3 | 7.2 | 27.1 | 5.9 | 1.2 | 5.1 | 0.8 | 4.3 | 0.9 | 2.4 | 2.3 | 0.4 | 2.3 | 1.0 | 26.1 | 13.9 | 3.0 |
| 149.3 | 143.9 | 81.0 | 290.0 | 27.6 | 78.0 | 13.3 | 0.6 | 3.2 | 2.5 | 8.3 | 851.1 | 28.7 | 63.0 | 6.9 | 26.2 | 5.8 | 1.3 | 5.4 | 0.8 | 4.8 | 1.0 | 2.6 | 2.5 | 0.4 | 2.2 | 0.9 | 30.1 | 12.0 | 2.1 |
| 9.6 | 47.1 | 104.1 | 130.4 | 34.2 | 31.6 | 15.1 | 0.3 | 3.9 | 0.5 | 5.7 | 316.0 | 72.3 | 118.3 | 14.8 | 53.6 | 10.4 | 1.5 | 8.2 | 1.2 | 6.2 | 1.2 | 3.1 | 2.9 | 0.5 | 1.0 | 1.3 | 16.6 | 30.8 | 4.1 |
| 51.0 | 109.3 | 181.9 | 134.3 | 25.6 | 60.4 | 18.7 | 0.4 | 6.1 | 1.1 | 15.6 | 547.2 | 33.0 | 79.4 | 8.0 | 28.9 | 6.0 | 1.2 | 5.3 | 0.8 | 4.5 | 0.9 | 2.5 | 2.4 | 0.4 | 1.8 | 1.4 | 41.5 | 18.1 | 3.1 |
| 226.1 | 143.9 | 113.7 | 635.9 | 25.6 | 67.4 | 11.1 | 0.4 | 2.9 | 1.2 | 7.7 | 647.7 | 30.0 | 59.5 | 6.9 | 25.4 | 5.4 | 1.2 | 5.0 | 0.8 | 4.3 | 0.9 | 2.4 | 2.3 | 0.4 | 1.8 | 0.8 | 31.9 | 11.6 | 2.0 |
| 44.5 | 96.2 | 124.6 | 543.8 | 17.8 | 63.4 | 13.6 | 1.2 | 3.1 | 1.6 | 8.3 | 430.8 | 29.0 | 60.9 | 6.8 | 25.3 | 5.0 | 1.0 | 4.1 | 0.6 | 3.3 | 0.7 | 1.8 | 1.6 | 0.2 | 1.8 | 0.9 | 24.9 | 13.2 | 3.9 |
| 42.8 | 103.3 | 150.8 | 201.7 | 19.1 | 65.8 | 14.5 | 1.8 | 3.8 | 1.1 | 10.4 | 576.5 | 33.4 | 72.7 | 7.8 | 28.5 | 5.5 | 1.1 | 4.5 | 0.7 | 3.7 | 0.7 | 1.9 | 1.8 | 0.3 | 1.9 | 1.0 | 23.9 | 16.4 | 3.7 |
| 44.7 | 84.1 | 97.1 | 1545.9 | 19.7 | 60.9 | 11.6 | 0.6 | 2.4 | 1.5 | 5.8 | 413.5 | 27.4 | 55.8 | 6.4 | 23.7 | 4.7 | 1.0 | 4.1 | 0.6 | 3.5 | 0.7 | 1.9 | 1.8 | 0.3 | 1.7 | 0.8 | 18.4 | 11.6 | 4.2 |
| 70.5 | 131.6 | 169.6 | 122.1 | 22.8 | 76.7 | 16.2 | 1.1 | 4.2 | 1.7 | 11.0 | 480.2 | 37.4 | 80.0 | 8.6 | 31.7 | 6.2 | 1.2 | 5.1 | 0.8 | 4.3 | 0.8 | 2.3 | 2.2 | 0.3 | 2.2 | 1.1 | 27.5 | 18.4 | 3.7 |
| 53.8 | 124.3 | 129.1 | 330.8 | 21.8 | 73.1 | 13.6 | 1.3 | 2.8 | 3.0 | 7.7 | 493.3 | 31.9 | 68.3 | 7.5 | 27.8 | 5.6 | 1.1 | 4.8 | 0.7 | 4.0 | 0.8 | 2.2 | 2.1 | 0.3 | 2.1 | 0.9 | 22.7 | 14.0 | 3.6 |
| 49.3 | 103.1 | 125.8 | 461.9 | 21.8 | 72.4 | 15.1 | 1.1 | 3.0 | 2.0 | 7.4 | 509.8 | 36.2 | 78.1 | 8.2 | 30.7 | 6.0 | 1.2 | 4.9 | 0.8 | 4.0 | 0.8 | 2.1 | 2.0 | 0.3 | 2.0 | 1.0 | 27.7 | 15.8 | 3.2 |
| 60.3 | 131.5 | 149.1 | 201.5 | 22.7 | 78.0 | 16.1 | 1.5 | 3.4 | 3.1 | 9.0 | 478.6 | 36.3 | 76.4 | 8.3 | 30.1 | 5.8 | 1.2 | 5.0 | 0.7 | 4.2 | 0.8 | 2.2 | 2.1 | 0.3 | 2.2 | 1.0 | 24.2 | 16.3 | 2.5 |
| 63.9 | 123.5 | 128.9 | 277.1 | 20.9 | 76.9 | 14.1 | 1.0 | 2.9 | 2.7 | 7.7 | 367.9 | 30.4 | 63.5 | 7.0 | 25.9 | 5.1 | 1.1 | 4.5 | 0.7 | 3.8 | 0.8 | 2.1 | 2.0 | 0.3 | 2.1 | 0.9 | 22.2 | 13.3 | 4.0 |
| 61.7 | 112.9 | 172.3 | 208.0 | 24.3 | 73.7 | 16.9 | 0.8 | 4.3 | 1.4 | 11.5 | 491.0 | 36.8 | 77.6 | 8.5 | 30.9 | 6.0 | 1.2 | 5.1 | 0.8 | 4.4 | 0.9 | 2.4 | 2.3 | 0.3 | 2.1 | 1.1 | 27.2 | 17.3 | 4.8 |
| 60.1 | 107.4 | 150.4 | 181.6 | 23.6 | 78.2 | 16.0 | 1.5 | 3.6 | 1.5 | 9.6 | 557.0 | 37.0 | 77.6 | 8.4 | 30.9 | 6.1 | 1.2 | 5.1 | 0.8 | 4.4 | 0.9 | 2.3 | 2.2 | 0.3 | 2.2 | 1.1 | 24.8 | 16.5 | 3.7 |
| 69.9 | 145.6 | 149.0 | 184.3 | 23.7 | 71.2 | 14.9 | 2.0 | 3.7 | 2.5 | 9.8 | 573.9 | 32.6 | 69.6 | 7.5 | 27.6 | 5.5 | 1.2 | 4.8 | 0.7 | 4.1 | 0.8 | 2.2 | 2.2 | 0.3 | 2.0 | 1.0 | 25.5 | 14.9 | 4.5 |
| 88.8 | 172.5 | 159.9 | 123.3 | 21.5 | 71.7 | 15.5 | 2.5 | 4.0 | 2.8 | 10.8 | 606.7 | 33.0 | 70.5 | 7.7 | 28.4 | 5.6 | 1.2 | 4.7 | 0.7 | 4.0 | 0.8 | 2.1 | 2.0 | 0.3 | 2.0 | 1.0 | 31.0 | 16.1 | 4.0 |
| 58.6 | 109.0 | 119.6 | 258.9 | 20.7 | 75.4 | 13.4 | 5.4 | 2.9 | 1.7 | 7.7 | 618.2 | 30.6 | 64.9 | 7.1 | 26.2 | 5.2 | 1.1 | 4.6 | 0.7 | 3.9 | 0.8 | 2.1 | 2.0 | 0.3 | 2.1 | 0.9 | 23.0 | 14.2 | 6.7 |
| 82.5 | 125.2 | 146.5 | 135.5 | 23.2 | 88.7 | 15.5 | 1.6 | 3.5 | 1.8 | 9.3 | 594.9 | 38.6 | 81.0 | 8.9 | 32.4 | 6.3 | 1.2 | 5.2 | 0.8 | 4.3 | 0.8 | 2.2 | 2.1 | 0.3 | 2.4 | 1.0 | 27.3 | 19.3 | 4.1 |
| 63.4 | 116.8 | 153.9 | 144.9 | 23.0 | 80.1 | 16.4 | 1.4 | 3.9 | 2.0 | 10.1 | 600.6 | 40.0 | 86.5 | 9.0 | 32.7 | 6.3 | 1.2 | 5.2 | 0.8 | 4.3 | 0.9 | 2.3 | 2.2 | 0.3 | 2.2 | 1.1 | 28.3 | 18.9 | 3.8 |
| 68.0 | 114.9 | 138.3 | 133.8 | 21.2 | 66.0 | 14.2 | 3.2 | 3.4 | 1.5 | 9.2 | 571.0 | 31.1 | 67.4 | 7.0 | 25.7 | 5.1 | 1.1 | 4.4 | 0.7 | 3.8 | 0.8 | 2.1 | 2.1 | 0.3 | 1.8 | 1.0 | 25.3 | 15.0 | 4.6 |
| 65.6 | 124.9 | 155.8 | 150.7 | 24.9 | 81.2 | 16.4 | 1.5 | 3.9 | 2.4 | 10.4 | 632.3 | 37.2 | 79.4 | 8.5 | 31.0 | 6.2 | 1.3 | 5.2 | 0.8 | 4.5 | 0.9 | 2.4 | 2.4 | 0.4 | 2.2 | 1.1 | 29.1 | 17.6 | 4.6 |
| 61.7 | 104.2 | 154.2 | 133.8 | 25.2 | 76.3 | 17.2 | 1.9 | 3.7 | 1.5 | 9.6 | 621.8 | 41.8 | 88.6 | 9.4 | 33.8 | 6.6 | 1.4 | 5.5 | 0.8 | 4.7 | 0.9 | 2.5 | 2.4 | 0.4 | 2.1 | 1.2 | 27.3 | 19.2 | 3.3 |
| 65.8 | 119.3 | 158.0 | 109.6 | 23.1 | 82.9 | 16.5 | 1.2 | 4.2 | 2.4 | 10.8 | 624.6 | 38.2 | 83.3 | 8.8 | 32.0 | 6.3 | 1.3 | 5.3 | 0.8 | 4.5 | 0.9 | 2.3 | 2.2 | 0.3 | 2.3 | 1.1 | 28.7 | 19.7 | 4.5 |
| 59.1 | 112.7 | 136.7 | 121.7 | 21.6 | 57.4 | 12.8 | 2.2 | 3.7 | 1.7 | 9.6 | 630.1 | 29.8 | 67.4 | 6.8 | 24.7 | 5.0 | 1.1 | 4.3 | 0.7 | 3.8 | 0.8 | 2.2 | 2.2 | 0.4 | 1.8 | 0.9 | 25.4 | 14.6 | 4.9 |
| 84.5 | 122.1 | 140.8 | 200.0 | 25.4 | 72.1 | 14.8 | 2.1 | 3.9 | 1.8 | 10.2 | 627.5 | 33.8 | 72.4 | 7.7 | 28.3 | 5.7 | 1.2 | 5.1 | 0.8 | 4.4 | 0.9 | 2.4 | 2.4 | 0.4 | 2.0 | 1.0 | 31.3 | 16.4 | 4.4 |
| 77.3 | 140.5 | 134.8 | 154.3 | 20.8 | 72.2 | 14.5 | 2.6 | 3.6 | 2.7 | 9.3 | 554.9 | 30.9 | 69.1 | 7.0 | 25.7 | 5.1 | 1.1 | 4.7 | 0.7 | 3.8 | 0.8 | 2.0 | 1.9 | 0.3 | 2.0 | 1.0 | 26.2 | 17.2 | 3.8 |
| 80.7 | 115.7 | 140.0 | 160.1 | 24.6 | 72.0 | 14.0 | 11.9 | 4.0 | 1.4 | 10.8 | 603.3 | 31.9 | 68.0 | 7.4 | 27.3 | 5.5 | 1.1 | 4.9 | 0.7 | 4.1 | 0.9 | 2.3 | 2.3 | 0.4 | 2.1 | 1.0 | 28.7 | 16.8 | 6.0 |
| 65.6 | 124.3 | 175.8 | 130.1 | 22.6 | 73.4 | 16.3 | 0.9 | 5.0 | 1.4 | 14.2 | 618.1 | 35.8 | 74.3 | 8.3 | 30.5 | 6.1 | 1.2 | 5.2 | 0.8 | 4.2 | 0.9 | 2.2 | 2.1 | 0.3 | 2.1 | 1.2 | 29.7 | 18.8 | 3.1 |
| 67.5 | 128.8 | 213.1 | 114.5 | 23.1 | 53.6 | 16.1 | 0.7 | 5.3 | 1.5 | 14.6 | 624.5 | 38.5 | 77.5 | 8.4 | 30.8 | 6.1 | 1.3 | 5.2 | 0.8 | 4.2 | 0.9 | 2.2 | 2.1 | 0.3 | 1.7 | 1.1 | 38.3 | 19.3 | 2.6 |
| 61.2 | 104.3 | 163.8 | 154.9 | 23.0 | 100.7 | 18.1 | 1.0 | 4.8 | 1.1 | 11.3 | 604.2 | 41.3 | 79.1 | 8.4 | 30.5 | 6.0 | 1.2 | 5.1 | 0.8 | 4.2 | 0.8 | 2.2 | 2.1 | 0.3 | 2.6 | 1.3 | 30.4 | 19.2 | 2.9 |
| 64.2 | 117.1 | 207.1 | 128.2 | 23.5 | 47.9 | 15.8 | 1.1 | 5.3 | 1.3 | 14.0 | 643.0 | 40.4 | 78.1 | 8.5 | 31.2 | 6.2 | 1.2 | 5.3 | 0.8 | 4.3 | 0.9 | 2.2 | 2.1 | 0.3 | 1.5 | 1.1 | 30.4 | 20.0 | 2.8 |
| 98.1 | 146.9 | 130.0 | 190.8 | 24.7 | 68.3 | 13.5 | 1.3 | 3.5 | 2.7 | 9.3 | 514.5 | 32.5 | 69.2 | 7.5 | 27.8 | 5.7 | 1.2 | 5.1 | 0.8 | 4.4 | 0.9 | 2.4 | 2.3 | 0.4 | 1.9 | 0.9 | 31.7 | 16.0 | 3.7 |
| 79.2 | 119.1 | 132.0 | 135.5 | 22.3 | 67.9 | 13.7 | 2.2 | 3.6 | 2.3 | 9.3 | 479.6 | 31.7 | 70.7 | 7.3 | 26.5 | 5.3 | 1.1 | 4.7 | 0.7 | 4.1 | 0.8 | 2.2 | 2.1 | 0.3 | 2.0 | 1.0 | 32.2 | 17.4 | 3.0 |
| 96.7 | 143.1 | 148.0 | 152.9 | 24.9 | 68.0 | 13.5 | 1.3 | 3.7 | 2.3 | 10.6 | 471.3 | 32.6 | 70.8 | 7.4 | 27.5 | 5.6 | 1.2 | 5.0 | 0.8 | 4.3 | 0.9 | 2.4 | 2.3 | 0.4 | 1.9 | 0.9 | 34.4 | 16.9 | 3.1 |

# Supplemental Table S2

| Exp | Site | Hole | Core | Type | Sect | W | Top | Bot | Depth | Age | CaCO$_3$ | Total Organic C | Total Carbon | Total N, acidified | δ$^{13}$C, TOC | Color |
|---|---|---|---|---|---|---|---|---|---|---|---|---|---|---|---|---|
| | | | | | | | | | mbsf, CSF-A | Ma | wt.% | wt.% | wt.% | wt.% | per mil | relative to pair |
| 353 | U1445 | A | 1 | H | 1 | W | 41 | 43 | 0.41 | 0.00 | 1.93 | 1.11 | 1.34 | 0.11 | -20.02 | Light |
| 353 | U1445 | A | 1 | H | 1 | W | 93 | 95 | 0.93 | 0.01 | 3.61 | 1.39 | 1.82 | 0.14 | -20.30 | Dark |
| 353 | U1445 | A | 1 | H | 5 | W | 23 | 25 | 6.23 | 0.04 | 26.81 | 1.64 | 4.86 | 0.16 | -17.23 | No Pair |
| 353 | U1445 | A | 4 | H | 1 | W | 100 | 102 | 26.9 | 0.20 | 11.04 | 1.59 | 2.91 | 0.16 | -17.90 | Dark |
| 353 | U1445 | A | 4 | H | 2 | W | 120 | 122 | 28.6 | 0.21 | 10.62 | 0.69 | 1.96 | 0.08 | -18.22 | Light |
| 353 | U1445 | A | 7 | H | 4 | W | 92 | 94 | 59.43 | 0.47 | 9.35 | 1.41 | 2.54 | 0.13 | -16.93 | Dark |
| 353 | U1445 | A | 7 | H | 4 | W | 112 | 114 | 59.63 | 0.47 | 7.02 | 1.15 | 1.99 | 0.12 | -17.79 | Light |
| 353 | U1445 | A | 10 | H | 1 | W | 95 | 97 | 83.85 | 0.68 | 2.88 | 0.94 | 1.29 | 0.12 | -19.23 | Light |
| 353 | U1445 | A | 10 | H | 3 | W | 41 | 43 | 85.88 | 0.70 | 5.42 | 1.51 | 2.16 | 0.13 | -18.98 | Dark |
| 353 | U1445 | A | 11 | H | 4 | W | 61 | 63 | 96.82 | 0.80 | 9.21 | 0.72 | 1.83 | 0.08 | -18.33 | No Pair |
| 353 | U1445 | A | 12 | H | 8 | W | 37 | 39 | 111.22 | 0.93 | 3.83 | 1.43 | 1.89 | 0.15 | -18.52 | Dark |
| 353 | U1445 | A | 13 | H | 1 | W | 88 | 90 | 112.28 | 0.94 | 4.09 | 1.08 | 1.57 | 0.12 | -18.56 | Light |
| 353 | U1445 | A | 16 | H | 1 | W | 70 | 72 | 140.6 | 1.22 | 3.07 | 1.02 | 1.38 | 0.11 | -18.31 | Light |
| 353 | U1445 | A | 16 | H | 2 | W | 65 | 67 | 141.81 | 1.23 | 1.72 | 1.38 | 1.58 | 0.16 | -19.28 | Dark |
| 353 | U1445 | A | 19 | H | 1 | W | 70 | 72 | 169.1 | 1.51 | 2.68 | 1.06 | 1.38 | 0.12 | -18.89 | Light |
| 353 | U1445 | A | 19 | H | 4 | W | 78 | 80 | 173.35 | 1.55 | 11.80 | 2.24 | 3.66 | 0.21 | -17.88 | Dark |
| 353 | U1445 | A | 22 | H | 1 | W | 30 | 32 | 197.2 | 1.86 | 2.39 | 1.84 | 2.12 | 0.19 | -19.67 | Light |
| 353 | U1445 | A | 22 | H | 2 | W | 93 | 95 | 199.33 | 1.89 | 4.86 | 1.90 | 2.48 | 0.21 | -18.19 | Dark |
| 353 | U1445 | A | 22 | H | 4 | W | 88 | 90 | 201.6 | 1.92 | 4.32 | 1.96 | 2.48 | 0.20 | -17.90 | No Pair |
| 353 | U1445 | A | 25 | X | 1 | W | 41 | 43 | 225.51 | 2.20 | 4.95 | 1.86 | 2.45 | 0.20 | -18.54 | Dark |
| 353 | U1445 | A | 25 | X | 3 | W | 12 | 14 | 227.5 | 2.22 | 5.33 | 1.15 | 1.79 | 0.12 | -17.13 | Light |
| 353 | U1445 | A | 28 | X | 1 | W | 104 | 106 | 255.24 | 2.42 | 3.83 | 2.16 | 2.62 | 0.22 | -17.39 | Dark |
| 353 | U1445 | A | 28 | X | 1 | W | 138 | 140 | 255.58 | 2.42 | 3.74 | 1.89 | 2.34 | 0.17 | -16.52 | Light |
| 353 | U1445 | A | 31 | X | 1 | W | 59 | 61 | 283.89 | 2.65 | 1.97 | 1.50 | 1.74 | 0.17 | -18.48 | Light |
| 353 | U1445 | A | 31 | X | 3 | W | 40 | 42 | 286.41 | 2.67 | 7.76 | 1.55 | 2.48 | 0.15 | -18.49 | Dark |
| 353 | U1445 | A | 34 | X | 1 | W | 19 | 21 | 310.89 | 2.89 | 13.81 | 2.08 | 3.73 | 0.20 | -19.18 | Light |
| 353 | U1445 | A | 34 | X | 2 | W | 53 | 55 | 312.64 | 2.91 | 1.98 | 2.60 | 2.84 | 0.23 | -18.63 | Dark |
| 353 | U1445 | A | 37 | X | 4 | W | 82 | 84 | 338.75 | 3.12 | 3.58 | 1.21 | 1.64 | 0.15 | -20.39 | Light |
| 353 | U1445 | A | 37 | X | 5 | W | 53 | 55 | 339.97 | 3.13 | 3.50 | 1.44 | 1.86 | 0.16 | -19.84 | Dark |
| 353 | U1445 | A | 41 | X | 1 | W | 117 | 119 | 368.57 | 3.35 | 3.29 | 2.73 | 3.13 | 0.23 | -19.09 | Light |
| 353 | U1445 | A | 41 | X | 2 | W | 11 | 13 | 368.93 | 3.35 | 3.63 | 3.32 | 3.76 | 0.26 | -18.51 | Dark |
| 353 | U1445 | A | 41 | X | 3 | W | 88 | 90 | 371.22 | 3.37 | | | | | | No Pair |
| 353 | U1445 | A | 44 | X | 4 | W | 61 | 63 | 396.06 | 3.53 | 1.57 | 0.61 | 0.80 | 0.08 | -19.83 | Light |
| 353 | U1445 | A | 44 | X | 4 | W | 136 | 138 | 396.81 | 3.53 | 3.88 | 1.01 | 1.48 | 0.12 | -20.15 | Dark |
| 353 | U1445 | A | 47 | X | 5 | W | 84 | 86 | 422.01 | 3.70 | 8.07 | 0.67 | 1.64 | 0.10 | -20.63 | Light |
| 353 | U1445 | A | 48 | X | 1 | W | 8 | 10 | 423.48 | 3.71 | 5.04 | 1.33 | 1.93 | 0.15 | -19.42 | Dark |
| 353 | U1445 | A | 51 | X | 4 | W | 74 | 76 | 451.55 | 3.87 | 6.88 | 0.80 | 1.62 | 0.10 | -21.47 | Dark |
| 353 | U1445 | A | 51 | X | 6 | W | 135 | 137 | 454.63 | 3.89 | 3.85 | 0.39 | 0.85 | 0.06 | -21.64 | Light |
| 353 | U1445 | A | 51 | X | 8 | W | 6 | 8 | 455.77 | 3.90 | 3.26 | 0.39 | 0.78 | 0.07 | -21.04 | No Pair |
| 353 | U1445 | A | 55 | X | 4 | W | 15 | 17 | 482.78 | 4.07 | 4.24 | 1.08 | 1.59 | 0.13 | -19.68 | Dark |
| 353 | U1445 | A | 55 | X | 5 | W | 69 | 71 | 484.82 | 4.08 | 3.52 | 0.39 | 0.81 | 0.06 | -19.74 | Light |
| 353 | U1445 | A | 59 | X | 2 | W | 57 | 59 | 512.37 | 4.27 | 3.80 | 1.11 | 1.57 | 0.14 | -19.79 | Dark |
| 353 | U1445 | A | 59 | X | 3 | W | 12 | 14 | 513.43 | 4.28 | 7.32 | 0.88 | 1.76 | 0.11 | -19.70 | Light |
| 353 | U1445 | A | 62 | X | 4 | W | 133 | 135 | 541.21 | 4.53 | 8.94 | 0.98 | 2.05 | 0.11 | -20.02 | Dark |
| 353 | U1445 | A | 62 | X | 5 | W | 68 | 70 | 542.06 | 4.53 | 4.47 | 0.42 | 0.96 | 0.07 | -20.49 | Light |
| 353 | U1445 | A | 66 | X | 1 | W | 106 | 108 | 568.46 | 4.84 | 4.17 | 0.90 | 1.40 | 0.11 | -19.92 | Dark |
| 353 | U1445 | A | 66 | X | 2 | W | 90 | 92 | 569.81 | 4.86 | 7.32 | 0.40 | 1.28 | 0.06 | -20.45 | Light |
| 353 | U1445 | A | 66 | X | 4 | W | 60 | 62 | 572.53 | 4.89 | 4.03 | 0.98 | 1.47 | 0.12 | -19.13 | No Pair |
| 353 | U1445 | A | 69 | X | 1 | W | 25 | 27 | 595.05 | 5.19 | 3.97 | 0.66 | 1.14 | 0.09 | -20.86 | Dark |
| 353 | U1445 | A | 69 | X | 1 | W | 148 | 150 | 596.28 | 5.20 | 3.88 | 0.39 | 0.85 | 0.07 | -21.24 | Light |
| 353 | U1445 | A | 71 | X | 7 | W | 63 | 65 | 623.54 | 5.59 | 6.71 | 0.78 | 1.58 | 0.09 | -20.97 | Light |
| 353 | U1445 | A | 72 | X | 1 | W | 90 | 92 | 624.8 | 5.61 | 9.26 | 0.93 | 2.04 | 0.11 | -19.61 | Dark |
| 353 | U1445 | A | 75 | X | 4 | W | 62 | 64 | 654.8 | 6.08 | 5.08 | 1.57 | 2.18 | 0.15 | -20.28 | Dark |
| 353 | U1445 | A | 75 | X | 4 | W | 121 | 123 | 655.39 | 6.09 | 1.91 | 0.93 | 1.16 | 0.11 | -20.05 | Light |
| 353 | U1445 | A | 76 | X | 4 | W | 79 | 81 | 664.72 | 6.25 | 3.35 | 1.38 | 1.78 | 0.15 | -20.57 | No Pair |
| 353 | U1445 | A | 76 | X | 5 | W | 28 | 29 | 665.33 | 6.26 | 5.19 | 0.54 | 1.16 | 0.08 | -21.92 | Light |
| 353 | U1445 | A | 76 | X | 5 | W | 73 | 75 | 665.78 | 6.26 | 3.66 | 0.72 | 1.16 | 0.09 | -21.42 | Dark |

**Supplemental Table S3**

| Exp | Site | Hole | Core | Type | Sect | W | Top | Bot | Depth | Age | δD, C26 | std δD, C26 | δD, C28 | std δD, C28 | δD, C30 | std δD, C30 | δ13C, C26 | std δ13C, C26 | δ13C, C28 | std δ13C, C28 | δ13C, C30 | std δ13C, C30 |
|---|---|---|---|---|---|---|---|---|---|---|---|---|---|---|---|---|---|---|---|---|---|---|
| | | | | | | | | | mbsf, CSF-A | Ma | ‰ | ‰ | ‰ | ‰ | ‰ | ‰ | ‰ | ‰ | ‰ | ‰ | ‰ | ‰ |
| 353 | U1445 | A | 1 | H | 1 | W | 41 | 43 | 0.41 | 0.00 | -151.73 | 0.47 | -157.68 | 1.98 | -156.62 | 1.03 | -26.72 | 0.07 | -28.09 | 0.07 | -28.11 | 0.02 |
| 353 | U1445 | A | 1 | H | 1 | W | 93 | 95 | 0.93 | 0.01 | -153.03 | 0.29 | -162.74 | 3.84 | -150.81 | 0.40 | -26.46 | 0.17 | -27.94 | 0.09 | -27.96 | 0.14 |
| 353 | U1445 | A | 1 | H | 5 | W | 23 | 25 | 6.23 | 0.04 | -141.93 | 1.86 | -143.37 | 3.13 | -141.05 | 1.00 | -23.37 | 0.05 | -22.42 | 0.13 | -22.45 | 0.13 |
| 353 | U1445 | A | 4 | H | 1 | W | 100 | 102 | 26.9 | 0.20 | -139.98 | 0.89 | -148.12 | 4.27 | -147.09 | 3.03 | -24.26 | 0.07 | -23.95 | 0.01 | -23.98 | 0.03 |
| 353 | U1445 | A | 4 | H | 2 | W | 120 | 122 | 28.6 | 0.21 | -133.87 | 1.55 | -146.27 | 4.68 | -131.69 | 2.10 | -24.04 | 0.10 | -23.87 | 0.13 | -23.90 | 0.02 |
| 353 | U1445 | A | 7 | H | 4 | W | 92 | 94 | 59.43 | 0.47 | -137.66 | 1.81 | -145.95 | 1.63 | -142.85 | 0.55 | -24.26 | 0.08 | -24.11 | 0.01 | -24.14 | 0.02 |
| 353 | U1445 | A | 7 | H | 4 | W | 112 | 114 | 59.63 | 0.47 | -130.70 | 2.28 | -140.82 | 2.17 | -131.13 | 3.42 | -23.25 | 0.02 | -23.52 | 0.08 | -23.55 | 0.00 |
| 353 | U1445 | A | 10 | H | 1 | W | 95 | 97 | 83.85 | 0.68 | -152.29 | 1.78 | -155.26 | 3.44 | -159.16 | 3.27 | -25.25 | 0.07 | -26.05 | 0.06 | -26.07 | 0.03 |
| 353 | U1445 | A | 10 | H | 3 | W | 41 | 43 | 85.88 | 0.70 | -148.73 | 1.49 | -155.95 | 1.96 | -149.26 | 6.34 | -26.26 | 0.04 | -26.73 | 0.09 | -26.76 | 0.04 |
| 353 | U1445 | A | 11 | H | 4 | W | 61 | 63 | 96.82 | 0.80 | -128.00 | 3.11 | -135.51 | 3.88 | -125.67 | 4.38 | -23.60 | 0.06 | -23.91 | 0.01 | -23.94 | 0.20 |
| 353 | U1445 | A | 12 | H | 8 | W | 37 | 39 | 111.22 | 0.93 | -149.03 | 1.31 | -158.62 | 1.53 | -164.55 | 1.48 | -25.05 | 0.03 | -24.97 | 0.07 | -25.00 | 0.02 |
| 353 | U1445 | A | 13 | H | 1 | W | 88 | 90 | 112.28 | 0.94 | -136.62 | 0.30 | -144.92 | 0.98 | -136.14 | 1.09 | -24.53 | 0.01 | -24.67 | 0.05 | -24.70 | 0.01 |
| 353 | U1445 | A | 16 | H | 1 | W | 70 | 72 | 140.6 | 1.22 | -142.15 | 1.61 | -153.11 | 1.28 | -150.62 | 0.56 | -25.08 | 0.08 | -25.28 | 0.06 | -25.30 | 0.05 |
| 353 | U1445 | A | 16 | H | 2 | W | 65 | 67 | 141.81 | 1.23 | -147.43 | 2.84 | -153.46 | 3.84 | -158.88 | 5.61 | -25.36 | 0.02 | -26.26 | 0.09 | -26.29 | 0.07 |
| 353 | U1445 | A | 19 | H | 1 | W | 70 | 72 | 169.1 | 1.51 | -148.96 | 2.27 | -153.25 | 2.80 | -160.43 | 2.75 | -25.76 | 0.02 | -27.05 | 0.04 | -27.07 | 0.07 |
| 353 | U1445 | A | 19 | H | 4 | W | 78 | 80 | 173.35 | 1.55 | -136.60 | 1.11 | -142.02 | 3.22 | -141.73 | 5.66 | -23.53 | 0.05 | -23.85 | 0.08 | -23.88 | 0.05 |
| 353 | U1445 | A | 22 | H | 1 | W | 30 | 32 | 197.2 | 1.86 | -141.35 | 2.29 | -145.42 | 2.00 | -155.34 | 7.42 | -24.16 | 0.03 | -25.03 | 0.01 | -25.05 | 0.05 |
| 353 | U1445 | A | 22 | H | 2 | W | 93 | 95 | 199.33 | 1.89 | -143.90 | 1.08 | -139.71 | 0.48 | -148.99 | 6.32 | -22.99 | 0.01 | -23.22 | 0.03 | -23.25 | 0.13 |
| 353 | U1445 | A | 22 | H | 4 | W | 88 | 90 | 201.6 | 1.92 | -145.68 | 1.83 | -143.63 | 0.09 | -155.25 | 0.89 | -22.88 | 0.07 | -23.13 | 0.05 | -23.16 | 0.06 |
| 353 | U1445 | A | 25 | X | 1 | W | 41 | 43 | 225.51 | 2.20 | -141.33 | 2.15 | -149.27 | 2.71 | -166.48 | 4.28 | -22.86 | 0.08 | -22.95 | 0.04 | -22.98 | 0.01 |
| 353 | U1445 | A | 25 | X | 3 | W | 12 | 14 | 227.5 | 2.22 | -144.88 | 2.91 | -147.08 | 2.77 | -149.81 | 3.50 | -23.16 | 0.18 | -23.59 | 0.03 | -23.62 | 0.01 |
| 353 | U1445 | A | 28 | X | 1 | W | 104 | 106 | 255.24 | 2.42 | -138.11 | 3.14 | -142.83 | 5.34 | -157.13 | 2.66 | -23.37 | 0.34 | -22.72 | 0.27 | -22.75 | 0.37 |
| 353 | U1445 | A | 28 | X | 1 | W | 138 | 140 | 255.58 | 2.42 | -149.31 | 1.56 | -151.96 | 2.61 | -159.70 | 2.09 | -22.81 | 0.10 | -22.42 | 0.14 | -22.45 | 0.20 |
| 353 | U1445 | A | 31 | X | 1 | W | 59 | 61 | 283.89 | 2.65 | -146.93 | 1.43 | -151.08 | 2.00 | -162.03 | 9.28 | -23.24 | 0.10 | -23.57 | 0.10 | -23.60 | 0.01 |
| 353 | U1445 | A | 31 | X | 3 | W | 40 | 42 | 286.41 | 2.67 | -145.32 | 2.10 | -148.70 | 2.12 | -147.90 | 11.51 | -25.22 | 0.12 | -25.04 | 0.10 | -25.07 | 0.03 |
| 353 | U1445 | A | 34 | X | 1 | W | 19 | 21 | 310.89 | 2.89 | -149.87 | 1.63 | -155.34 | 1.67 | -158.31 | 0.38 | -25.05 | 0.11 | -25.56 | 0.19 | -25.59 | 0.10 |
| 353 | U1445 | A | 34 | X | 2 | W | 53 | 55 | 312.64 | 2.91 | -145.28 | 3.31 | -153.61 | 0.88 | -159.28 | 3.47 | -23.65 | 0.14 | -24.14 | 0.05 | -24.17 | 0.04 |
| 353 | U1445 | A | 37 | X | 4 | W | 82 | 84 | 338.75 | 3.12 | -149.87 | 3.48 | -155.06 | 0.77 | -162.80 | 1.78 | -23.56 | 0.06 | -25.12 | 0.08 | -25.14 | 0.07 |
| 353 | U1445 | A | 37 | X | 5 | W | 53 | 55 | 339.97 | 3.13 | -151.54 | 4.50 | -160.89 | 2.58 | -170.93 | 2.70 | -24.32 | 0.03 | -25.48 | 0.05 | -25.50 | 0.11 |
| 353 | U1445 | A | 41 | X | 1 | W | 117 | 119 | 368.57 | 3.35 | -130.77 | 2.33 | -151.26 | 0.46 | -158.58 | 0.97 | -22.52 | 0.05 | -23.62 | 0.04 | -23.65 | 0.02 |
| 353 | U1445 | A | 41 | X | 2 | W | 11 | 13 | 368.93 | 3.35 | -140.24 | 0.67 | -157.94 | 1.43 | -153.53 | 1.58 | -23.14 | 0.07 | -24.12 | 0.04 | -24.15 | 0.02 |
| 353 | U1445 | A | 41 | X | 3 | W | 88 | 90 | 371.22 | 3.37 | -145.85 | 0.58 | -147.95 | 0.70 | -153.74 | 1.40 | -23.14 | 0.04 | -24.07 | 0.04 | -24.10 | 0.07 |
| 353 | U1445 | A | 44 | X | 4 | W | 61 | 63 | 396.06 | 3.53 | | | | | | | -25.55 | 0.04 | -26.25 | 0.05 | -26.27 | 0.02 |
| 353 | U1445 | A | 44 | X | 4 | W | 136 | 138 | 396.81 | 3.53 | -140.30 | 2.98 | -146.67 | 4.56 | -156.81 | 2.19 | -25.16 | 0.03 | -26.57 | 0.01 | -26.59 | 0.01 |
| 353 | U1445 | A | 47 | X | 5 | W | 84 | 86 | 422.01 | 3.70 | -152.00 | 1.45 | -158.51 | 2.48 | -169.11 | 3.31 | -26.06 | 0.10 | -27.90 | 0.10 | -27.92 | 0.17 |
| 353 | U1445 | A | 48 | X | 1 | W | 8 | 10 | 423.48 | 3.71 | -139.20 | 2.12 | -150.46 | 0.62 | -160.79 | 2.02 | -25.59 | 0.26 | -26.12 | 0.09 | -26.14 | 0.20 |
| 353 | U1445 | A | 51 | X | 4 | W | 74 | 76 | 451.55 | 3.87 | -165.27 | 0.27 | -166.48 | 1.81 | -174.20 | 2.84 | -26.67 | 0.01 | -28.52 | 0.01 | -28.54 | 0.03 |
| 353 | U1445 | A | 51 | X | 6 | W | 135 | 137 | 454.63 | 3.89 | -167.57 | 14.07 | -166.00 | 13.27 | -165.73 | 19.03 | | | | | | |
| 353 | U1445 | A | 51 | X | 8 | W | 6 | 8 | 455.77 | 3.90 | -165.28 | 1.10 | -164.92 | 3.11 | -167.31 | 3.11 | | | | | | |
| 353 | U1445 | A | 55 | X | 4 | W | 15 | 17 | 482.78 | 4.07 | -143.20 | 1.73 | -146.78 | 0.82 | -154.59 | 1.89 | -24.63 | 0.10 | -26.06 | 0.06 | -26.08 | 0.11 |
| 353 | U1445 | A | 55 | X | 5 | W | 69 | 71 | 484.82 | 4.08 | -143.99 | 2.88 | -150.25 | 0.44 | -147.29 | 10.15 | | | | | | |
| 353 | U1445 | A | 59 | X | 2 | W | 57 | 59 | 512.37 | 4.27 | -148.85 | 1.68 | -153.56 | 0.41 | -160.90 | 0.94 | | | | | | |
| 353 | U1445 | A | 59 | X | 3 | W | 12 | 14 | 513.43 | 4.28 | -144.53 | 1.79 | -150.71 | 3.93 | -153.98 | 1.67 | -23.96 | 0.09 | -25.37 | 0.04 | -25.40 | 0.25 |
| 353 | U1445 | A | 62 | X | 4 | W | 133 | 135 | 541.21 | 4.53 | -150.33 | 2.70 | -157.17 | 1.35 | -163.73 | 2.45 | -24.67 | 0.20 | -25.38 | 0.09 | -25.40 | 0.21 |
| 353 | U1445 | A | 62 | X | 5 | W | 68 | 70 | 542.06 | 4.53 | -163.90 | 0.32 | -172.06 | 0.35 | -172.96 | 2.48 | | | | | | |
| 353 | U1445 | A | 66 | X | 1 | W | 106 | 108 | 568.46 | 4.84 | -148.84 | 1.54 | -155.75 | 1.73 | -156.34 | 3.24 | -23.71 | 0.05 | -24.64 | 0.11 | -24.67 | 0.07 |
| 353 | U1445 | A | 66 | X | 2 | W | 90 | 92 | 569.81 | 4.86 | -159.16 | 2.82 | -173.90 | 2.17 | -179.29 | 3.23 | | | | | | |
| 353 | U1445 | A | 66 | X | 4 | W | 60 | 62 | 572.53 | 4.89 | -136.66 | 5.40 | -147.46 | 1.63 | -148.37 | 1.60 | -23.11 | 0.03 | -24.29 | 0.03 | -24.32 | 0.02 |
| 353 | U1445 | A | 69 | X | 1 | W | 25 | 27 | 595.05 | 5.19 | -152.54 | 0.59 | -154.11 | 1.14 | -154.99 | 1.51 | -24.27 | 0.07 | -27.12 | 0.12 | -27.14 | 0.19 |
| 353 | U1445 | A | 69 | X | 1 | W | 148 | 150 | 596.28 | 5.20 | -169.76 | 0.90 | -174.34 | 2.31 | -173.60 | 2.02 | | | | | | |
| 353 | U1445 | A | 71 | X | 7 | W | 63 | 65 | 623.54 | 5.59 | -159.87 | 0.48 | -164.74 | 3.41 | -168.14 | 2.88 | -26.43 | 0.14 | -27.20 | 0.00 | -27.23 | 0.05 |
| 353 | U1445 | A | 72 | X | 1 | W | 90 | 92 | 624.8 | 5.61 | -148.30 | 2.22 | -157.45 | 0.66 | -163.92 | 3.54 | -24.30 | 0.02 | -24.25 | 0.08 | -24.28 | 0.01 |
| 353 | U1445 | A | 75 | X | 4 | W | 62 | 64 | 654.8 | 6.08 | -147.43 | 2.92 | -154.77 | 3.26 | -161.95 | 0.56 | -24.36 | 0.17 | -24.73 | 0.19 | -24.75 | 0.08 |
| 353 | U1445 | A | 75 | X | 4 | W | 121 | 123 | 655.39 | 6.08 | -156.62 | 2.43 | -161.57 | 1.27 | -174.29 | 1.94 | -24.87 | 0.07 | -24.64 | 0.02 | -24.67 | 0.05 |
| 353 | U1445 | A | 76 | X | 4 | W | 79 | 81 | 664.72 | 6.25 | -151.91 | 1.60 | -154.79 | 1.42 | -161.89 | 3.20 | -25.27 | 0.06 | -25.51 | 0.02 | -25.54 | 0.09 |
| 353 | U1445 | A | 76 | X | 5 | W | 28 | 29 | 665.33 | 6.26 | -154.48 | 3.70 | -164.14 | 3.46 | -165.42 | 0.64 | -26.13 | 0.14 | -27.19 | 0.21 | -27.21 | 0.08 |
| 353 | U1445 | A | 76 | X | 5 | W | 73 | 75 | 665.78 | 6.26 | -147.26 | 1.05 | -156.95 | 2.15 | -157.90 | 1.77 | -27.92 | 0.01 | -29.17 | 0.07 | -29.18 | 0.06 |