# Peer review of "Pliocene expansion of C4 vegetation in the core monsoon zone on the Indian Peninsula"

_Climate of the Past, 2020_

## Referee Comment (RC1) · Anonymous Referee #1 · 26 Apr 2020

Dunlea, Giosan and Huang use lipid plant wax biomarkers and their isotope composition to reconstruct the climate and vegetation of the eastern Indian peninsula over the last 6 million year. They find that C4 vegetation was already present in the late Miocene and further expanded during the Pliocene. They argue that this expansion was likely caused by changing precipitation patterns during the studied time period as well as a decline in atmospheric CO2 during the Pliocene. The results are consistent with previous reconstructions from the region that yielded similar results. The manuscript is overall well written. As outlined below there are however shortcomingh regardingh the description of the methodology that should be addressed prior to its publication and several points where the authors should expand the manuscript and provide further information.

Major comments: The authors should consider moving the method description from the appendix to the main text to have it in chronological order. In the journal format of Climate of the Past there is no reason to put the methodology at the end.

Lines 82-85: As is, it is confusing to see sample pairs mentioned without proper explanation. I would suggest do give a detailed description of the core and the reasoning behind the sampling strategy beforehand to avoid any confusion.

In Fig. 2C and lines 113-115 it is mentioned that the dD values are corrected for physiological effects of C3 and C4 photosynthesis. Unfortunately, there is no description available of how this correction was conducted. This should be added in order to be able to reproduce the calculations and presented results.

In lines 147-153, the Pliocene C4 expansion is explained by a lowering in atmospheric $CO_2$. It would be useful to show the $CO_2$ and C4 vegetation trends together in a figure to illustrate this point.

In the description of the compound-specific isotope analysis it is mentioned that the fatty acids were methylated prior to analysis (i.e. a methyl group added). Since this methyl group changes the isotope composition of the resulting fatty acid methyl esters, the measurements need to be corrected using the isotope composition of the methanol used in the reaction. Without the proper correction the absolute values and associated interpretations are incorrect.

Minor comments: Line 27: Add some references backing up the sentence ending at the beginning of this line.

Line 47: Specify what kind of model was used in the cited study.

Lines 92-94: Why are the mid-Pliocene (3-5 Myr) and mid-Pleistocene (1.5 Myr) selected in this description. On the figure, the trend in dD seems to be pretty constant and the selection of these time points seems rather arbitrary.

Lines 113-114: This sentence on physiological effects of C3 versus C4 plants on dD

is not really connected to the previous sentence on airmass mixing. It is therefore confusing to see the word thus at the beginning of this sentence.

Line 182: The unit cm3 already implies volume. The word volume after the unit is therefore redundant and can be deleted.

Line 210: Provide the isotope composition of the methanol used.

Lines 225: Specify the standard used. Was it an industry standard with known isotope values? Figures: In the text the abbreviation for million years is Myr, while in the figures Ma is used. This should be homogenized.

In the method description, alkenone and alkane measurements are described which are however not mentioned in the rest of the manuscript. Of course, it would be interesting to see these results. So, the authors should either remove reference to these measurements or include them in the manuscripts.

Supplementary tables S1-S3: In order to facilitate the use of data by other scientists, consider moving the data contained in the tables to a separate file that is in a machine-readable format.

---

## Referee Comment (RC2) · Anonymous Referee #2 · 24 Jun 2020

Dunlea et al. evaluate new organic geochemical data from the Bay of Bengal (Indian Ocean) in order to investigate changes in terrestrial biomes over the past 6 Ma. Specifically, the carry out: (i) geochemical analyses of major, trace and rare elements on sediments samples from IODP Site U1445 to determine the sediment provenance; (ii) $\delta 13C$ and $\delta D$ analysis on leaf-wax fatty acids (C30) to reconstruct the evolution of C4 plants since the late Miocene. The authors suggest that the sediment originates from the Mahanadi River in core monsoon area of the Indian Peninsula, and hence the biomarker data allow to better understand the timing of the C4 plant expansion during the Late Cenozoic in this region. They conclude that although C4 plants have been growing on the Indian Peninsula already during the late Miocene, they expanded strongly from the mid-Pliocene (at c. 3.5 Ma) onwards in agreement with previous

observations from East Africa and NW Australia.

The authors have produced a nice dataset that merits publication in Climate of the Past in principle. In its current form, however, the manuscript is poorly structured with most of the important information provided in the supplementary files. It also lacks an in-depth discussion of the results, particularly comparisons with other existing records from both marine and terrestrial settings and an assessment of the potential mechanisms behind the expansion of C4 plants during the Pliocene. These are dealt with in such a superficial detail that an interested reader from the broad audience of Climate of the Past who knows little about C4 plants and their expansion in the Late Cenozoic, would struggle to follow the arguments. If the authors are interested in greatly expanding the manuscript then I would support acceptance after rewriting.

When revising their manuscript, the authors should carefully address the following points:

1. More information on the age model development is needed. How reliable are the magneto- and biostratigraphic tie points used? Were the turbidite layers removed before developing the age model? What are the sedimentation rates, and how do they change through time? It is difficult to imagine that the sedimentation rates stay 'fairly constant' for a such a long time as the authors argue in line 68.

2. Please explain what you mean with higher latitudes and elevations in line 152. Reconstructions of C3/C4 vegetation in the Chinese Loess Plateau (An et al. 2005) and palynological records from the Tibetan Plateau (Koutsodendris et al. 2019) – which are arguably from higher latitudes and elevations than the study area – show expansion of C4 plants and arid semi-desert biomes, respectively, during the mid-Pliocene; hence the argument that ecosystems at higher latitudes and elevations remained stable is not correct. By extension, the interpretation that tropical ecosystems adjacent to the Indian Ocean are more sensitive and the $CO_2$ change is likely not the primary driver of the Pliocene C4 expansion is not fully substantiated.

3. The early Pleistocene interval (c. 2 - 1 Ma) is characterized by lower $\delta$13C values suggesting contraction of C4 vegetation in the study area. Please elaborate on this issue in a revised manuscript. Is a similar pattern also observed in other records? What kind of mechanism could be responsible?

4. The discussion on the global patterns of C4 expansion should be substantially expanded. Please also consider including recently published biomarker data from the western Indian Ocean (e.g., Pollisar et al., 2019) and also comparing the data from Site U1445 with palynological records from adjacent regions to the Indian Ocean (e.g., Miao et al., 2017; Koutsodendris et al., 2019) that also span the time interval from the mid-Miocene to Pleistocene.

5. The influence of precipitation as a trigger for the C4 expansion during the mid-Pliocene is also poorly explained (lines 154-155). The authors simply list several climate components affecting the precipitation variability in the Indian Ocean today without however explaining how they may have influenced the hydroclimate during the mid-Pliocene. They should at least elaborate on whether these climate systems were active during this time interval based on proxy records and model studies, and suggest specific mechanisms responsible for the hydroclimate, and in turn, vegetation dynamics in the study region.

6. The quality of plots is generally poor and it is difficult to evaluate the proxy records (a prime example is the $\delta$13C record from Site 231 in Fig. 3c). Please redraw the figures to increase clarity.

References: An, Z., Huang, Y., Liu, W., Guo, Z., Clemens, S., Li, L., Prell, W., Ning, Y., Cai, Y., Zhou, W., Lin, B., Zhang, Q., Cao, Y., Qiang, X., Chang, H., Wu, Z., 2005. Multiple expansions of C4 plant biomass in East Asia since 7 Ma coupled with strengthened monsoon circulation. Geology 33, 705.

Koutsodendris, A., Allstädt, F.J., Kern, O.A., Kousis, I., Schwarz, F., Vannacci, M., Woutersen, A., Appel, E., Berke, M.A., Fang, X., Friedrich, O., Hoorn, C., Salzmann, U.,

Pross, J., 2019. Late Pliocene vegetation turnover on the NE Tibetan Plateau (Central Asia) triggered by early Northern Hemisphere glaciation. Global and Planetary Change 180, 117-125.

Miao, Y., Warny, S., Clift, P.D., Liu, C., Gregory, M., 2017. Evidence of continuous Asian summer monsoon weakening as a response to global cooling over the last 8 Ma. Gondwana Research 52, 48-58.

Polissar, P.J., Rose, C., Uno, K.T., Phelps, S. R., deMenocal, P., 2019. Synchronous rise of African $C_4$ ecosystems 10 million years ago in the absence of aridification. Nature Geoscience, 12, 657-660.
* * *

---

## Author Response (AR1)

*Author's Response to Reviews of* "Pliocene expansion of $C_4$ vegetation in the core monsoon zone on the Indian Peninsula" *by* Ann G. Dunlea et al.

**Response to Editor Comments**

Editor Decision: Reconsider after major revisions (05 Aug 2020) by **_Alberto Reyes_**
Comments to the Author:

Dear Dr. Dunlea,

Thank you for submitting responses to the comments and suggestions of the two reviewers. Both reviewers are positive about the dataset and the underlying science questions. There is also broad consensus from the reviewers that the presentation of methods and discussion can be bulked up, either through additional text or re-organization. Regarding Rev. 2's comment about the age model, I think it should be possible to provide the requested information on age model construction while referring readers to the published source(s) for full details on the tie-points used in your age model.

Your responses indicate that you are willing to undertake these revisions, so I invite you to submit a revised manuscript for further consideration. The revised manuscript will go through another stage of peer-review, which requires that I indicate "major revisions".

Sincerely,
Alberto Reyes (handling editor)

- Dear Dr. Alberto Reyes,

  Thank you for your comments on our manuscript. Below you can find a point-by-point response to the reviewer's comments and the corresponding revised manuscript with changes tracked. Substantial revisions of the manuscript draft have been made, including an expansion of the discussion section and other edits raised by reviewers. Please let me know if you have any questions regarding the revisions.

  Sincerely,
  Ann Dunlea

**Response to Anonymous Referee #1**

Dunlea, Giosan and Huang use lipid plant wax biomarkers and their isotope composition to reconstruct the climate and vegetation of the eastern Indian peninsula over the last 6 million year. They find that C4 vegetation was already

present in the late Miocene and further expanded during the Pliocene. They argue that this expansion was likely caused by changing precipitation patterns during the studied time period as well as a decline in atmospheric CO2 during the Pliocene. The results are consistent with previous reconstructions from the region that yielded similar results. The manuscript is overall well written. As outlined below there are however shortcomings regarding the description of the methodology that should be addressed prior to its publication and several points where the authors should expand the manuscript and provide further information.

- We thank the anonymous referee for their comments and revisions.

Major comments: The authors should consider moving the method description from the appendix to the main text to have it in chronological order. In the journal format of Climate of the Past there is no reason to put the methodology at the end.

- The methods section has been moved from the appendix to the main text.

Lines 82-85: As is, it is confusing to see sample pairs mentioned without proper explanation. I would suggest do give a detailed description of the core and the reasoning behind the sampling strategy beforehand to avoid any confusion.

- A better description of the sampling pairs was added to the methods.

In Fig. 2C and lines 113-115 it is mentioned that the dD values are corrected for physiological effects of C3 and C4 photosynthesis. Unfortunately, there is no description available of how this correction was conducted. This should be added in order to be able to reproduce the calculations and presented results.

- We added the explanation of the correction to the methods section.

In lines 147-153, the Pliocene C4 expansion is explained by a lowering in atmospheric CO2. It would be useful to show the CO2 and C4 vegetation trends together in a figure to illustrate this point.

- Estimates of atmospheric $CO_2$ have been added to Figure 3.

In the description of the compound-specific isotope analysis it is mentioned that the fatty acids were methylated prior to analysis (i.e. a methyl group added). Since this methyl group changes the isotope composition of the resulting fatty acid methyl esters, the measurements need to be corrected using the isotope composition of the methanol used in the reaction. Without the proper correction the absolute values and associated interpretations are incorrect.

- The correction was performed and we added a description of the correction to the methods section.

Minor comments: Line 27: Add some references backing up the sentence ending at the beginning of this line.

- The following references were added to support the statement: An et al., 2005; Behrensmeyer et al., 2007; Huang et al., 2007; Edwards et al., 2010; Zhou et al., 2014

Line 47: Specify what kind of model was used in the cited study.

- We added details about the type of model used to predict natural flora was added. The reference cited has additional information.

Lines 92-94: Why are the mid-Pliocene (3-5 Myr) and mid-Pleistocene (1.5 Myr) selected in this description. On the figure, the trend in dD seems to be pretty constant and the selection of these time points seems rather arbitrary.

- We selected those time intervals to be consistent with the time intervals we used for the $\delta^{13}C$ plots. We added text to emphasize the change in $\delta D$ is gradually increasing and we only remark on the time interval to compare with the $\delta^{13}C$ record.

Lines 113-114: This sentence on physiological effects of C3 versus C4 plants on dD is not really connected to the previous sentence on airmass mixing. It is therefore confusing to see the word thus at the beginning of this sentence.

- The sentence was edited for better logical flow and includes reference to the methods section that explains the correction for plant physiology.

Line 182: The unit cm3 already implies volume. The word volume after the unit is therefore redundant and can be deleted.

- The redundancy was removed.

Line 210: Provide the isotope composition of the methanol used.

- We have added the isotopic composition of the methanol used and the equations used to correct the data for the addition of methanol.

Lines 225: Specify the standard used. Was it an industry standard with known isotope values?

- Done. It was an in-house lab standard with known values.

Figures: In the text the abbreviation for million years is Myr, while in the figures Ma is used. This should be homogenized.

- We are using "Myr" for "million years", and "Ma" for "million years ago", as the USGS does. "Ma" thus stands for an event in the past whereas "Myr" indicates duration or time interval. The text was checked for consistency of this notation.

In the method description, alkenone and alkane measurements are described which are however not mentioned in the rest of the manuscript. Of course, it would be interesting to see these results. So, the authors should either remove reference to these measurements or include them in the manuscripts.

- We removed the reference to the alkenone and alkane measurements.

Supplementary tables S1-S3: In order to facilitate the use of data by other scientists, consider moving the data contained in the tables to a separate file that is in a machine- readable format.

- Yes, we will submit the final tables as .xls or .csv files.

**Response to Anonymous Referee #2**

Dunlea et al. evaluate new organic geochemical data from the Bay of Bengal (Indian Ocean) in order to investigate changes in terrestrial biomes over the past 6 Ma. Specifically, the carry out: (i) geochemical analyses of major, trace and rare elements on sediments samples from IODP Site U1445 to determine the sediment provenance; (ii) 13C and D analysis on leaf-wax fatty acids (C30) to reconstruct the evolution of C4 plants since the late Miocene. The authors suggest that the sediment originates from the Mahanadi River in core monsoon area of the Indian Peninsula, and hence the biomarker data allow to better understand the timing of the C4 plant expansion during the Late Cenozoic in this region. They conclude that although C4 plants have been growing on the Indian Peninsula already during the late Miocene, they expanded strongly from the mid-Pliocene (at c. 3.5 Ma) onwards in agreement with previous observations from East Africa and NW Australia.

The authors have produced a nice dataset that merits publication in Climate of the Past in principle. In its current form, however, the manuscript is poorly structured with most of the important information provided in the supplementary files. It also lacks an in-depth discussion of the results, particularly comparisons with other existing records from both marine and terrestrial settings and an assessment of the potential mechanisms behind the expansion of C4 plants during the Pliocene. These are dealt with in such a superficial detail that an interested reader from the broad audience of Climate of the Past who knows little about C4 plants and their expansion in the Late Cenozoic, would struggle to follow the arguments. If the authors are interested in greatly expanding the manuscript then I would support acceptance after rewriting.

- We thank the reviewer for their comments and suggested revisions. Substantial portions of the discussion have been re-written.

When revising their manuscript, the authors should carefully address the following points:

1. More information on the age model development is needed. How reliable are the magneto- and biostratigraphic tie points used? Were the turbidite layers removed be- fore developing the age model? What are the sedimentation rates, and how do they change through time? It is difficult to imagine that the sedimentation rates stay 'fairly constant' for a such a long time as the authors argue in line 68.

- We added the average sedimentation rate with uncertainty to the main text. We agree that the age model is a critical part of any paleoceanographic study, but the age points we use are published and improving them is not the focus of this study. The line we fit to the previously published age constraints was merely to interpolate the ages to our samples specifically. While we agree that there may have been shorter-term changes in sedimentation rates, Supplemental Figure S1 demonstrates that there are no major or abrupt changes in sedimentation rate over the million year timescales of interest to this study.

2. Please explain what you mean with higher latitudes and elevations in line 152. Reconstructions of C3/C4 vegetation in the Chinese Loess Plateau (An et al. 2005) and palynological records from the Tibetan Plateau (Koutsodendris et al. 2019) – which are arguably from higher latitudes and elevations than the study area – show expansion of C4 plants and arid semi-desert biomes, respectively, during the mid-Pliocene; hence the argument that ecosystems at higher latitudes and elevations remained stable is not correct. By extension, the interpretation that tropical ecosystems adjacent to the Indian Ocean are more sensitive and the CO2 change is likely not the primary driver of the Pliocene C4 expansion is not fully substantiated.

- Yes, we have clarified this point. We were saying that the Indian Peninsula is more sensitive to monsoon changes than the relatively close records in the India, Pakistan, and the Himalayan regions. We added the Pliocene C4 patterns in Asia to the discussion.

3. The early Pleistocene interval (c. 2 - 1 Ma) is characterized by lower 13C values suggesting contraction of C4 vegetation in the study area. Please elaborate on this issue in a revised manuscript. Is a similar pattern also observed in other records? What kind of mechanism could be responsible?

- We added a paragraph discussing the change in the early Pleistocene.

4. The discussion on the global patterns of C4 expansion should be substantially expanded. Please also consider including recently published biomarker data from the western Indian Ocean (e.g., Pollisar et al., 2019) and also comparing the data from Site U1445 with palynological records from adjacent regions to the Indian Ocean (e.g., Miao et al., 2017; Koutsodendris et al., 2019) that also span the time interval from the mid-Miocene to Pleistocene.

- Thank you for these additional references. We included them in the expanded discussion.

5. The influence of precipitation as a trigger for the C4 expansion during the mid-Pliocene is also poorly explained (lines 154-155). The authors simply list several climate components affecting the precipitation variability in the Indian Ocean today without however explaining how they may have influenced the hydroclimate during the mid-Pliocene. They should at least elaborate on whether these climate systems were active during this time interval based on proxy records and model studies, and suggest specific mechanisms responsible for the hydroclimate, and in turn, vegetation dynamics in the study region.

- The explanation of the climate system and possible mechanisms has been expanded.

6. The quality of plots is generally poor and it is difficult to evaluate the proxy records (a prime example is the 13C record from Site 231 in Fig. 3c). Please redraw the figures to increase clarity.

- We have revised the plots to increase aesthetic value and clarity.

References: An, Z., Huang, Y., Liu, W., Guo, Z., Clemens, S., Li, L., Prell, W., Ning, Y., Cai, Y., Zhou, W., Lin, B., Zhang, Q., Cao, Y., Qiang, X., Chang, H., Wu, Z., 2005. Multiple expansions of C4 plant biomass in East Asia since 7 Ma coupled with strengthened monsoon circulation. Geology 33, 705.

Koutsodendris, A., Allstädt, F.J., Kern, O.A., Kousis, I., Schwarz, F., Vannacci, M., Woutersen, A., Appel, E., Berke, M.A., Fang, X., Friedrich, O., Hoorn, C., Salzmann, U., Pross, J., 2019. Late Pliocene vegetation turnover on the NE Tibetan Plateau (Central Asia) triggered by early Northern Hemisphere glaciation. Global and Planetary Change 180, 117-125.

Miao, Y., Warny, S., Clift, P.D., Liu, C., Gregory, M., 2017. Evidence of continuous Asian summer monsoon weakening as a response to global cooling over the last 8 Ma. Gondwana Research 52, 48-58.

Polissar, P.J., Rose, C., Uno, K.T., Phelps, S. R., deMenocal, P., 2019. Synchronous rise of African C 4 ecosystems 10 million years ago in the absence of aridification. Nature Geoscience, 12, 657-660.

**Pliocene expansion of C4 vegetation in the core monsoon zone on the Indian Peninsula**

Ann G. Dunlea[1], Liviu Giosan[2], Yongsong Huang[3]

[1]Marine Chemistry & Geochemistry, Woods Hole Oceanographic Institution, Woods Hole, MA, 02543, USA
[2]Geology & Geophysics, Woods Hole Oceanographic Institution, Woods Hole, MA, 02543, USA
[3]Department of Earth, Environmental, and Planetary Sciences, Brown University, Providence, RI, 02912, USA

*Correspondence to*: Ann G. Dunlea (adunlea@whoi.edu)

**Abstract.** The expansion of C4 vegetation during the Neogene was one of the largest reorganizations of Earth's terrestrial biome. Once thought to be globally synchronous in the late Miocene, site-specific studies have revealed differences in the timing of the expansion and suggest that local conditions play a substantial role. Here, we examine the expansion of C4 vegetation on the Indian Peninsula since the late Miocene by constructing a ~6 million year paleorecord with marine sediment from the Bay of Bengal at Site U1445 drilled during International Ocean Discovery Program Expedition 353. Analyses of element concentrations indicate the marine sediment originates from the Mahanadi River in the Core Monsoon Zone (CMZ) of the Indian Peninsula. Hydrogen isotopes of the fatty acids of leaf waxes reveal an overall decrease in the CMZ precipitation since the late Miocene. Carbon isotopes of the leaf wax fatty acids suggest C4 vegetation on the Indian Peninsula existed before the end of the Miocene, but expanded to even higher abundances during the mid-Pliocene to mid-Pleistocene (~3.5 to 1.5 million years ago). Similar to the CMZ on the Indian Peninsula, a Pliocene expansion or re-expansion has previously been observed in northwest Australia and in East Africa, suggesting that these tropical ecosystems surrounding the Indian Ocean remained highly sensitive to changes in hydroclimate after the initial spread of C4 plants in late Miocene.

| | Deleted: ( |
| | Deleted: Ma |
| | Deleted: climate |

**1. Introduction**

[revised manuscript text omitted]